# Catching the Details: Self-Distilled RoI Predictors for Fine-Grained MLLM Perception

**Yuheng Shi**[1]    **Xiaohuan Pei**[1]    **Minjing Dong**[2]    **Chang Xu**[1]
[1]University of Sydney    [2]City University of Hong Kong
`yshi0087@uni.sydney.edu.au, xiaohuan.pei@sydney.edu.au,`
`minjdong@cityu.edu.hk, c.xu@sydney.edu.au`

## ABSTRACT

Multimodal Large Language Models (MLLMs) require high-resolution visual information to perform fine-grained perception, yet processing entire high-resolution images is computationally prohibitive. While recent methods leverage a Region-of-Interest (RoI) mechanism to focus on salient areas, they typically present a difficult trade-off: training-based approaches depend on large-scale annotated datasets, while training-free methods that utilize the model's internal attention are computationally inefficient and less accurate, requiring either multi-pass prefill stages or reliance on the slow auto-regressive decoding process. In this paper, we propose an efficient, annotation-free **S**elf-**D**istilled **R**egion **P**roposal Network (SD-RPN) that resolves this trade-off. The SD-RPN is built around a pipeline that transforms the noisy attention maps from the MLLM's middle layers into high-quality pseudo-RoI labels by explicitly denoising the signal and resolving ambiguity. We use these labels to train a Region Proposal Network (RPN) that learns a more precise localization. This RPN is also highly efficient, predicting the RoI in a single forward pass using features from the MLLM's middle layers, decoupling RoI identification from the auto-regressive generation and avoiding costly multi-pass operations. To validate our approach, we integrate the framework into the LLaVA-1.5 architecture. Despite being trained on only a few (*e.g.* 10K) question-answer pairs, our method demonstrates exceptional data efficiency and generalization, achieving over a 10% absolute accuracy improvement on unseen benchmarks, including TextVQA, DocVQA, and V-Star. Our work presents a practical and scalable solution for enhancing the fine-grained perception of MLLMs without requiring costly supervision or full model fine-tuning. Code is available at `https://github.com/YuHengsss/SD-RPN`.

## 1 Introduction

Recent years have witnessed significant advancements in Multimodal Large Language Models (MLLMs), which have evolved from foundational architectures like LLaVA (Liu et al., 2023b) to more sophisticated systems (Wang et al., 2024a; Bai et al., 2023; Chen et al., 2024a;b) such as Qwen2.5-VL (Bai et al., 2025) and InternVL-3.0 (Zhu et al., 2025). The performance of MLLMs is tied to the quality of their visual perception. In a typical MLLM architecture, a vision encoder, such as CLIP (Radford et al., 2021), processes visual signals and projects them into the embedding space of a Large Language Model (LLM) (Touvron et al., 2023; Zheng et al., 2023) for subsequent reasoning. Consequently, the richness of these visual features is crucial for the model to achieve a comprehensive and fine-grained understanding of the input.

To enhance this fine-grained perception, scaling the resolution of visual inputs has emerged as a direct and effective strategy. The approaches to achieve this have evolved considerably. Initial methods (Liu et al., 2023b; Dai et al., 2023; Bai et al., 2023) relied on a fixed-resolution input. Subsequent research introduced more flexible techniques, such as $S^2$ (Shi et al., 2024), Any-Resolution (Liu et al., 2023a; Chen et al., 2024b), and Naive Dynamic Resolution (Wang et al., 2024a; Bai et al., 2025) to handle higher-resolution imagery. However, processing entire high-resolution images uniformly is computationally intensive. More recently, an alternative paradigm (Yu et al., 2025; Shi

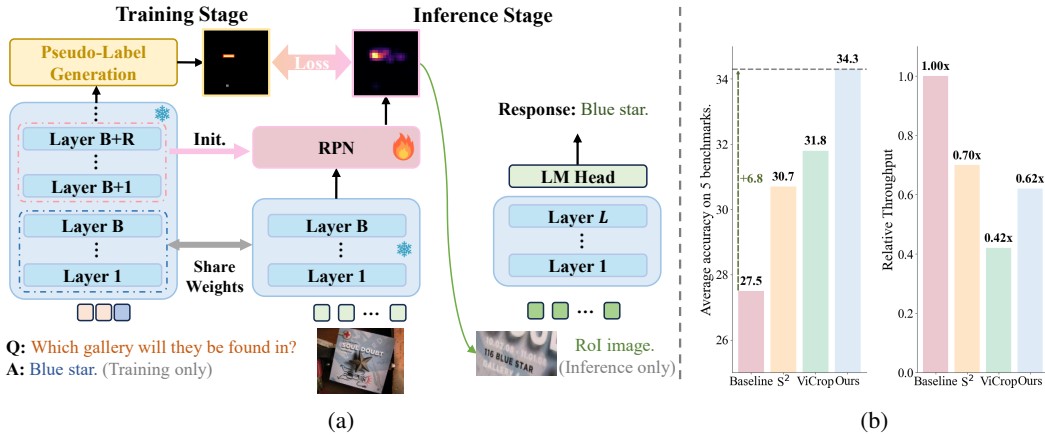

Figure 1: **(a) The Pipeline of SD-RPN**. The RPN is trained with pseudo-labels to effectively predict RoIs. These RoIs are then used to crop fine-grained sub-images for the final inference stage. **(b) Performance Comparison**. Performance evaluation with $S^2$ (Shi et al., 2024) and ViCrop (Zhang et al., 2025a) on the LLaVA-1.5-7B baseline. Accuracy is averaged over five Document and OCR benchmarks.

et al., 2025a) has gained traction: identifying a Region-of-Interest (RoI) within a low-resolution input and then selectively scaling up only that specific region. This RoI-centric approach has proven to be a more efficient and effective means of improving visual detail perception.

While the RoI paradigm marks a significant step toward efficient high-resolution processing, current methodologies for identifying these regions still present notable limitations. Approaches such as VILA-HD (Shi et al., 2025a), for instance, rely on large-scale pre-training with detailed annotations, a process that is both data-intensive and computationally demanding. Furthermore, they often require a complete prefilling stage for the initial low-resolution image, which can impede both training and inference efficiency. An alternative direction leverages the intrinsic localization capabilities of MLLMs (Kang et al., 2025b; Zhang et al., 2025a) without training, specifically by computing cross-attention scores between image tokens and the corresponding textual description. However, leveraging this internal attention for RoI identification is often computationally inefficient in inference, as current methods typically require either complex, multi-pass operations during the prefill stage (Zhang et al., 2025a) or rely on the inherently sequential and slow auto-regressive decoding stage (Wang et al., 2024b). Consequently, an effective and efficient method for RoI identification that avoids both reliance on extensive annotated data and the high latency of auto-regressive decoding or multiple forward passes remains a critical, yet underexplored, challenge.

In this work, we introduce SD-RPN, a novel self-distillation framework (as shown in Fig. 1a) designed to overcome these limitations by efficiently harnessing the MLLM's intrinsic localization capabilities. Our approach is motivated by the insight that while an MLLM's internal attention provides a strong RoI signal, it is too noisy (e.g., attention sinks, incomplete activation) for direct supervision. Previous studies reveal that using such noisy signals for dense supervision yields suboptimal results (Wang et al., 2022; Tian et al., 2021). To address this, we propose a pseudo-labeling pipeline that transforms the noisy attention map into a sparse and reliable supervisory signal. This pipeline first denoises the map by removing sink tokens based on feature norms. Subsequently, it employs a selective classification strategy that assigns discrete labels based on confidence thresholds and a minimal bounding box around high-attention tokens, which resolves the ambiguity inherent in the original noisy map. We use these pseudo labels to train a RPN. By learning from this distilled knowledge, the RPN develops a more precise localization function than methods relying on raw, ambiguous attention. Beyond its accuracy, the RPN's architectural design confers significant efficiency gains. Composed of a few transformer blocks built upon the frozen MLLM backbone, the RPN operates on features from the model's middle layers. This strategic placement allows it to predict the RoI by executing only a partial forward pass up to the MLLM's middle layers, completely decoupling localization from the slow, auto-regressive generation process. The entire framework is trained end-to-end, distilling the localization knowledge from the model's own response-guided attention into the efficient RPN. This enables a dynamic two-stage inference process where the model first predicts salient regions and then analyzes high-resolution crops to generate its final response.

Our SD-RPN achieves a significant enhancement (Fig. 1b) in fine-grained perception without the cost of full-model fine-tuning or large-scale annotated datasets. Moreover, our framework demonstrates exceptional training efficiency and generalization. Using only 10K random samples from GQA (Hudson & Manning, 2019) and OCR-VQA (Mishra et al., 2019) within the LLaVA-1.5 (Liu et al., 2023a) framework, our approach yields substantial gains on unseen data, boosting accuracy by over 10% on TextVQA (Singh et al., 2019), DocVQA (Mathew et al., 2020), and V-Star (Wu & Xie, 2023). We further confirm the robustness of our method by applying it to the more recent DeepSeek-VL (Lu et al., 2024) and Qwen2.5-VL (Bai et al., 2025) architectures. Our contributions are twofold. First, we introduce a robust pipeline to denoise the internal attention maps of an MLLM, generating high-quality pseudo-labels for supervision. Second, we propose a novel, annotation-free self-distillation framework that trains a RPN to predict RoIs by leveraging the MLLM's intrinsic localization knowledge.

## 2 RELATED WORKS

### 2.1 PERCEPTION IN MLLMS.

Recent studies have established that MLLMs often struggle with fine-grained perception, a limitation rooted in the challenge of efficiently processing high-resolution visual inputs (Tong et al., 2024b; Kang et al., 2025b; Shi et al., 2024; Liu et al., 2024a; Chen et al., 2024a). One line of research has focused on enhancing the model's global visual understanding. This includes developing more sophisticated vision encoders (Wang et al., 2024a; Bai et al., 2025; Zhang et al., 2024b; Ge et al., 2024; Luo et al., 2024), supplementing the LLM with full high-resolution images (Liu et al., 2024a; Zhu et al., 2025; Huang et al., 2024; Tong et al., 2024a; Cha et al., 2024; Li et al., 2025; 2024b), and incorporating external tools (Zhao et al., 2024). More recent works Shi et al. (2025a); Yu et al. (2025); Zheng et al. (2025); Shao et al. (2024) have demonstrated that identifying the RoI first in relatively low-resolution visual input and then scaling the resolution specifically for the RoI is more efficient and effective. However, they come at the expense of extensive training, requiring massive supervision and costly annotations. While recent training-free methods (Zhang et al., 2025a; Wang et al., 2024b; Shi et al., 2025b) attempt to identify RoIs by utilizing the internal perceptual capabilities of VLMs, they are often hampered by noisy activations or require slow auto-regresssive decoding stage and multiple forward passes, which hinders both performance and efficiency.

### 2.2 SELF-DISTILLATION IN MLLMS.

Knowledge distillation is an effective paradigm for transferring knowledge from a teacher to a student network (Hinton et al., 2015). Among its variants, self-distillation (Zhang et al., 2019) emerges as a unique form in which the teacher and student share the same architecture, and it has been widely adopted in vision–language tasks. In multimodal pre-training, several works (Oquab et al., 2023; Cai et al., 2024; Dong et al., 2023; Kim et al., 2025) leverage self-distillation to improve cross-modal alignment and representation learning, where it is commonly employed to enhance feature extraction. In multimodal downstream tasks, recent efforts (Kong et al., 2024; Peng et al., 2025; Hou et al., 2024) take advantage of self-distillation to refine vision–text region alignment and thereby improve grounding and reasoning performance. As self-distillation eliminates the need for a larger pre-trained teacher, it naturally offers strong potential for extracting high-resolution and fine-grained perceptual cues directly from MLLMs themselves.

## 3 METHOD

### 3.1 PRELIMINARIES

The standard architecture of a Multimodal Large Language Model (MLLM) comprises three core components: a vision encoder, $\mathcal{E}_v$, a vision-language projector, $\mathcal{P}$, and a Large Language Model (LLM), $\mathcal{L}$. Given an image-text pair, $(x_v, x_t)$, the MLLM processes the inputs to generate a textual response. The vision encoder $\mathcal{E}_v$ extracts feature vectors from the input image $x_v$. These visual features are then transformed by the projector $\mathcal{P}$ into a sequence of visual embeddings, $\mathbf{H}_v^0 = \mathcal{P}(\mathcal{E}_v(x_v))$, which are aligned with the LLM's input space. The input text $x_t$ is processed by a tokenizer and an embedding layer to produce a sequence of text embeddings, which

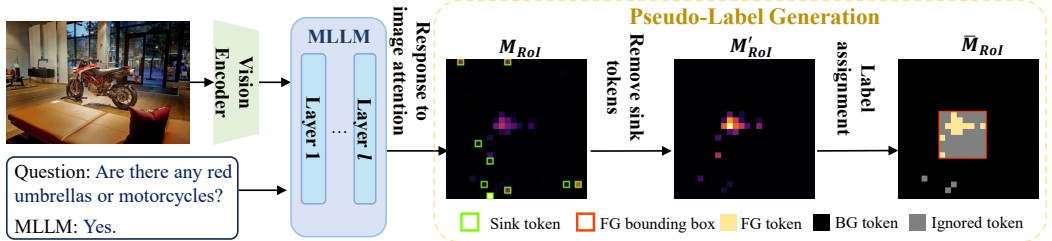

Figure 2: **An overview of our pseudo-label generation pipeline.** FG and BG denote the foreground and background respectively. Layer index is omitted for simplicity.

typically includes system prompt and user query, yielding $\mathbf{H}^0_{sys}$ and $\mathbf{H}^0_{user}$, respectively. After encoding $\mathbf{H}^0_{sys}, \mathbf{H}^0_v, \mathbf{H}^0_{user}$ in parallel (called the prefilling stage), the MLLM generates responses auto-regressively (called the decoding stage). Due to the lack of parallelization, the decoding stage is typically much slower than the prefilling stage when processing the same number of tokens.

The attention scores are key to identifying image regions relevant to the text (Kang et al., 2025b). From the cross-modal attention in layer $l$, we could derive a RoI map, $\mathbf{M}^l_{\text{RoI}} \in \mathbb{R}^{H \times W}$, which represents the averaged importance of each visual token across textual tokens of user query or response. For a single attention head, this map is computed as $\mathbf{M}^l_{\text{RoI}} = \sum_{i=1}^{N_t} \mathbf{A}^l_i / N_t$, $\mathbf{A} = \text{softmax}\left(\mathbf{Q}^l_t (\mathbf{K}^l_v)^T / \sqrt{d}\right)$, where $\mathbf{Q}^l_t \in \mathbb{R}^{N_t \times d}$ and $\mathbf{K}^l_v \in \mathbb{R}^{(H \times W) \times d}$ are the query and key matrices from the $N_t$ response tokens and $H \times W$ visual tokens, respectively.

## 3.2 PSEUDO-LABEL GENERATION FOR ROI

While the RoI map $\mathbf{M}_{\text{RoI}}$ provides a valuable signal for localizing text-relevant image regions, it is often fraught with noise. As illustrated in Fig. 2, these raw maps can exhibit erroneously high attention in background areas and incomplete activation across the true foreground object. To overcome this, we propose a pseudo-label generation pipeline, which is depicted in Fig. 2, to transform the noisy RoI map into a sparse and reliable supervisory signal for training our RPN.

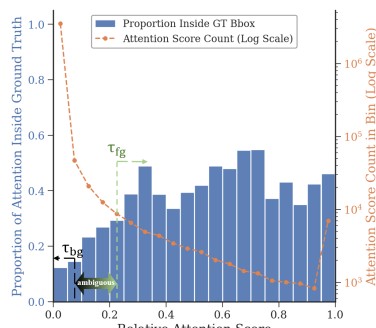

Figure 3: **Attention magnitude VS. Localization accuracy**.

The first source of noise we address is the sink tokens. These are visual tokens that attract substantial attention despite being semantically irrelevant to the grounded object. As identified in recent studies (Darcet et al., 2023; Kang et al., 2025a), sink tokens can be identified by the high L2-norm of their corresponding feature vectors $\mathbf{H}_v$. To mitigate them, we first denoise the initial RoI map $\mathbf{M}_{\text{RoI}}$ suppressing sink tokens identified via a predefined norm threshold $\tau_{\text{norm}}$, yielding a cleaner version, $\mathbf{M}'_{\text{RoI}}$:

$$(\mathbf{M}'_{\text{RoI}})_j = \begin{cases} 0 & \text{if } ||(\mathbf{H}_v)_j||_2 > \tau_{\text{norm}} \\ (\mathbf{M}_{\text{RoI}})_j & \text{otherwise} \end{cases}. \tag{1}$$

**Label Assignment.** Following the removal of sink tokens, the denoised RoI map $\mathbf{M}'_{\text{RoI}}$ more clearly highlights the foreground area. However, it is not yet an ideal supervisory signal, as it still suffers from an obscure foreground-background margin and incomplete object activation. To empirically investigate the margin issue, we analyze the denoised attention maps $\mathbf{M}'_{\text{RoI}}$ on the TextVQA subset (Zhang et al., 2025a). Fig. 3 plots the proportion of tokens falling inside the ground-truth (GT) bounding box as a function of their sample-wise relative attention score, calculated as $a/a_{\text{max}}$ for each attention value $a \in \mathbf{M}'_{\text{RoI}}$. The histogram reveals that the proportion of tokens inside the GT box is approximately 40% for high attention scores (e.g., $> 0.2$) and 10% for low scores (e.g., $< 0.1$). However, the ambiguous middle-range of attention not only exhibit a distinct localization pattern from the more decisive high and low scores, but they also vastly outnumber the high-attention tokens. Using such a signal for dense supervision would inevitably force the model to learn from these numerous ambiguous regions. To address this, we avoid regressing on $\mathbf{M}'_{\text{RoI}}$ directly and instead implement a selective binary classification that assigns foreground or background only to

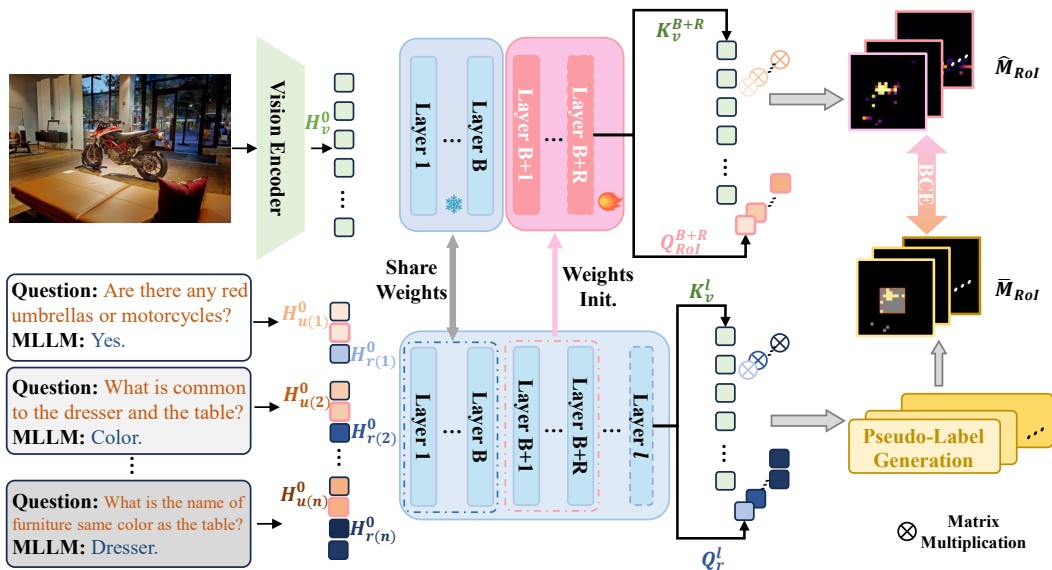

Figure 4: **Overview of our SD–RPN framework.** Our RPN (top) is initialized from and built upon a frozen MLLM backbone to efficiently predict a dense RoI map. It is trained via self-distillation (bottom), where pseudo-labels are generated by denoising the full MLLM's internal response-to-image attention maps. Superscripts denote layer indices; subscripts denote token sources. We omit the system prompt tokens for brevity.

high-confidence tokens, leaving ambiguous tokens to be ignored. This approach is also designed to alleviate the incomplete activation problem. We define a minimal bounding box, $\mathcal{B}_{fg}$, which encloses identified foreground tokens. Tokens inside this box that is not classified as foreground is explicitly ignored, which prevents the RPN from receiving erroneous background signals from inactivated parts of the true object. Guided by these criteria, the final pseudo-label map, $\bar{\mathbf{M}}_{\text{RoI}}$ is constructed by:

$$(\bar{\mathbf{M}}_{\text{RoI}})_j = \begin{cases} 1 & \text{if token } j \in \mathcal{S}_{fg}, \\ 0 & \text{if token } j \in \mathcal{S}_{bg}, \\ -1 & \text{otherwise (ignored)}, \end{cases} \quad (2)$$

where the foreground set is $\mathcal{S}_{fg} = \{j \mid a_j \geq \tau_{fg} a_{\max}\}$ and the background set is $\mathcal{S}_{bg} = \{j \mid j \notin \mathcal{B}_{fg} \text{ and } a_j \leq \tau_{bg} a_{\max}\}$. In Appendix F, we also provide a theoretical view of why learning to predict RoI labels is superior than using the raw attention maps directly.

### 3.3 RoI Prediction via Self-distillation

As validated by recent studies (Kang et al., 2025b; Shao et al., 2025), the middle layers of MLLMs exhibit significant potential for RoI prediction. Given these observations, can we leverage the localization capabilities inherent in the middle layers of MLLMs to build a more efficient RoI predictor in both inference and data? To answer this question, we propose a tunable Self-Distilled Region Proposal Network (SD-RPN) that consists of $R$ transformer blocks built upon the first $B$ frozen MLLM layers, which serve as the backbone.

**Predicting the RoI Map.** The architecture of our SD-RPN is illustrated in Fig. 4. We initialize the RPN's weights using those from layers $B$ to $B + R$ of the pretrained MLLM, a strategy that enables the efficient transfer of learned representations (Shao et al., 2025). Instead of predicting sparse bounding boxes, our well-initialized RPN is trained to predict a dense RoI map, $\hat{\mathbf{M}}_{\text{RoI}}$. The prediction process is triggered by specific query tokens derived from the conversational context. Concretely, for a given image and a conversation with $n$ turns, we first collect the sequence of hidden states, $\mathbf{H}$, from the second last layer of the RPN:

$$\mathbf{H} = [\mathbf{H}_{\text{sys}}, \mathbf{H}_v, \mathbf{H}_{u(1)}, \mathbf{H}_{r(1)}, \dots, \mathbf{H}_{u(n)}, \mathbf{H}_{r(n)}], \quad (3)$$

where the subscript of $u$ and $r$ denote user and response tokens. From this sequence, we specifically isolate the hidden state corresponding to the final token of each user question, $\mathbf{H}_{u(i)}[-1]$, as these

tokens immediately precede the model's answers and serve as the most direct prompt for generating a grounded response. These $n$ query vectors are collected into a single tensor, $\mathbf{H}_{\text{RoI}}$, which serves as the input for our RoI prediction head:

$$\mathbf{H}_{\text{RoI}} = \text{concat}(\mathbf{H}_{u(1)}[-1], \ldots, \mathbf{H}_{u(n)}[-1]), \tag{4}$$

where $\mathbf{H}_{\text{RoI}} \in \mathbb{R}^{n \times d}$. These query vectors and token hidden states $\mathbf{H}_v$, are then projected into the query and key space using the linear layers ($LP_q$ and $LP_k$) from the RPN's final attention block:

$$\mathbf{Q}_{\text{RoI}} = LP_q(\text{Norm}(\mathbf{H}_{\text{RoI}})), \quad \mathbf{K}_v = LP_k(\text{Norm}(\mathbf{H}_v)), \tag{5}$$

where Norm denotes the layer normalization (Zhang & Sennrich, 2019; Ba et al., 2016). The predicted dense RoI map, $\hat{\mathbf{M}}_{\text{RoI}}$ is then computed via a simple matrix multiplication of these query and key matrices: $\hat{\mathbf{M}}_{\text{RoI}} = \mathbf{Q}_{\text{RoI}}\mathbf{K}_v^T$. For brevity, the head dimension is omitted here; in practice, attention scores are computed per head and then averaged. Notably, this design is more efficient than exsiting frameworks (Shi et al., 2025a; Zhang et al., 2025a) as it only need a single forward through partial LLM layers.

**Training via Self-Distillation.** The RPN is trained entirely through self-distillation to predict the pseudo-label map $\bar{\mathbf{M}}_{\text{RoI}}$ which is generated via the pipeline in Section 3.2 by minimizing a binary cross-entropy (BCE) loss, defined as $\mathcal{L}_{\text{BCE}}(\hat{\mathbf{M}}_{\text{RoI}}, \bar{\mathbf{M}}_{\text{RoI}})$. While the text responses used for pseudo-label generation could theoretically originate from a stronger teacher model (*e.g.*, GPT-4) or human annotations, our empirical results reveal a crucial insight: using the MLLM's self-predicted responses yields superior performance (detailed in Tab. 4b). We attribute this counter-intuitive result to the principle of representational consistency. Although an external teacher's response may be more accurate, the attention maps it induces can be out-of-distribution for the student RPN. In contrast, the attention maps from self-generated responses, even if imperfect, are inherently aligned with its own internal visual grounding mechanisms. This creates a more consistent and attainable distillation target, especially in our data-efficient setting. This self-sufficient framework thus removes any dependency on external models or annotated data.

## 3.4 Two-stage inference with RoI

The predicted RoI map, $\hat{\mathbf{M}}_{\text{RoI}}$, enables a dynamic, two-stage inference process that significantly enhances the model's fine-grained perception capabilities. The first stage involves predicting and post-processing $\hat{\mathbf{M}}_{\text{RoI}}$ to produce a clean binary foreground mask, $\mathcal{B}$. To consolidate activated regions and ensure robustness against noise, the dense map $\hat{\mathbf{M}}_{\text{RoI}}$ is reshaped into a 2D map ($\gamma$), smoothed with a Gaussian filter ($\mathcal{G}$), and then binarized using a fixed threshold ($\tau$):

$$\mathcal{B}(x, y) = \begin{cases} 1, & \text{if } \mathcal{G}(\gamma(\hat{\mathbf{M}}_{\text{RoI}}))(x, y) > \tau, \\ 0, & \text{otherwise,} \end{cases} \tag{6}$$

where $(x, y)$ represents the spatial coordinates. With this binary mask, we proceed to the second stage to extract fine-grained visual features. We explore two different upscaling strategies for the predicted RoI. The first, which we term Box Upscaling, processes each salient region independently. We first identify all distinct connected-component regions, $\{\mathcal{R}^i\}_{i=1}^k$, within the mask $\mathcal{B}$. A minimal, axis-aligned bounding box, $b_i$, is then computed for each region. These bounding boxes are used to crop sub-images, $\{x_{v_i}\}_{i=1}^k$, from the original source image, which are then encoded to produce new, high-resolution visual embeddings $\mathbf{H}_{v_{\text{box}}}^0$:

$$b_i = \text{bbox}(\mathcal{R}_i), \quad \mathbf{H}_{v_{\text{box}}}^0 = \{\mathcal{P}(\mathcal{E}_v(x_{v_i}))\}_{i=1}^k, \tag{7}$$

where $\text{bbox}(\cdot)$ is the operator that returns the coordinates of the minimal bounding box enclosing the input region. Alternatively, our second strategy, which we term Masked Upscaling, takes a unified approach. This method first computes a single, all-encompassing bounding box, $b_{\text{all}}$, that encloses the union of all connected foreground regions, $\bigcup_{i=1}^k \mathcal{R}_i$. This unified bounding box is used to crop a single sub-image, $x_{v_{\text{all}}}$, which is then passed through the vision encoder and projector to produce a set of high-resolution embeddings $\mathbf{H}_{v_{\text{mask}}}^0$:

$$b_{\text{all}} = \text{bbox}(\bigcup_{i=1}^k \mathcal{R}_i), \quad \mathbf{H}_{v_{\text{mask}}}^0 = \mathcal{P}(\mathcal{B} \circ \mathcal{E}_v(x_{v_{\text{all}}})), \tag{8}$$

Table 1: Performance on Document & OCR benchmarks. Dataset subscripts denote the evaluation split. Performance subscripts show the absolute improvement (↑) over the baseline. Throughput is relative to the baseline, measured on a single NVIDIA A6000 GPU.

| Methods | Throughput | DocVQA$_{val}$ | ChartQA$_{test}$ | OCRBench$_{test}$ | InfoVQA$_{val}$ | TextVQA$_{val}$ | Ave. |
|---|---|---|---|---|---|---|---|
| **LLaVA-1.5-7B** | **1.0**× | 21.5 | 18.1 | 31.4 | 20.4 | 46.1 | 27.5 |
| +S$^2$ | 0.70× | 27.1 | 18.9 | 32.6 | **22.5** | 52.6 | 30.7 |
| +ViCrop | 0.42× | 27.0 | 20.0 | 33.2 | 21.4 | 57.2 | 31.8 |
| +SD-RPN(Ours) | 0.62× | **34.2**$_{↑12.7}$ | **20.6**$_{↑2.5}$ | **37.3**$_{↑5.9}$ | 22.3$_{↑1.9}$ | **58.8**$_{↑12.7}$ | **34.6**$_{↑7.1}$ |
| **LLaVA-1.5-13B** | **1.0**× | 23.5 | 18.1 | 33.7 | 23.4 | 48.7 | 29.5 |
| +S$^2$ | 0.71× | 30.7 | 20.3 | 36.4 | 24.7 | 54.5 | 33.3 |
| +ViCrop | 0.39× | 30.2 | 20.1 | 36.1 | **25.9** | 60.3 | 34.5 |
| +SD-RPN(Ours) | 0.51× | **39.4**$_{↑15.9}$ | **21.2**$_{↑3.1}$ | **39.6**$_{↑5.9}$ | 24.8$_{↑1.4}$ | **63.4**$_{↑14.5}$ | **37.7**$_{↑8.2}$ |
| **DeepSeek-VL-1.3B** | **1.0**× | 37.0 | 48.6 | 38.9 | 21.7 | 55.8 | 40.4 |
| +SD-RPN(Ours) | 0.47× | **54.4**$_{↑17.4}$ | **51.7**$_{↑3.1}$ | **40.8**$_{↑1.9}$ | **24.1**$_{↑2.4}$ | **65.9**$_{↑10.1}$ | **47.4**$_{↑7.0}$ |
| **DeepSeek-VL-7B** | **1.0**× | 50.7 | 61.4 | 42.4 | 30.4 | 63.0 | 49.6 |
| +SD-RPN(Ours) | 0.40× | **67.2**$_{↑16.5}$ | **64.2**$_{↑2.8}$ | **47.9**$_{↑5.5}$ | **36.8**$_{↑6.4}$ | **71.5**$_{↑8.5}$ | **57.5**$_{↑7.9}$ |
| **Qwen2.5-VL-3B** | **1.0**× | 87.1 | 82.6 | 73.8 | 62.7 | 76.9 | 76.6 |
| +SD-RPN(Ours) | 0.56× | **89.5**$_{↑2.4}$ | **83.8**$_{↑1.2}$ | **76.3**$_{↑2.5}$ | **67.0**$_{↑4.3}$ | **79.7**$_{↑2.8}$ | **79.3**$_{↑2.7}$ |
| **Qwen2.5-VL-7B** | **1.0**× | 92.0 | 82.6 | 81.5 | 70.2 | 81.1 | 81.5 |
| +SD-RPN(Ours) | 0.57× | **93.6**$_{↑1.6}$ | **85.5**$_{↑2.9}$ | **82.9**$_{↑1.4}$ | **76.9**$_{↑6.7}$ | **83.5**$_{↑2.4}$ | **84.5**$_{↑3.0}$ |

Table 2: Performance on Vision-Centric and High-Resolution benchmarks. † denotes results reported in the original publication.

| Methods | V* Bench | | | POPE | | HR-Bench 4K | | | HR-Bench 8K | | |
|---|---|---|---|---|---|---|---|---|---|---|---|
| | Attr | Spatial | Overall | F1 | Acc. | FSP | FCP | Overall | FSP | FCP | Overall |
| **LLaVA-1.5-7B** | 48.7 | 52.6 | 50.3 | 85.9 | 87.4 | 39.5 | 35.5 | 37.5 | 32.5 | 33.8 | 33.8 |
| +S$^2$ | 53.0 | 59.1 | 55.5 | 87.1 | 87.9 | 49.8 | **38.3** | 44.0 | 40.5 | **36.5** | 38.5 |
| +ViCrop | 53.9 | 50.0 | 52.4 | **88.0** | **88.6** | **60.8** | 34.8 | **47.8** | 38.5 | 33.8 | 36.1 |
| +SD-RPN(Ours) | **70.4** | **71.1** | **70.7**$_{↑20.4}$ | 87.2$_{↑1.4}$ | 87.9$_{↑0.5}$ | 59.0 | 35.5 | 47.3$_{↑9.8}$ | **48.8** | 34.5 | **41.6**$_{↑7.8}$ |
| **LLaVA-1.5-13B** | 47.0 | 56.6 | 50.8 | 85.9 | 87.1 | 41.5 | 44.5 | 43.0 | 36.6 | 38.5 | 36.6 |
| +S$^2$ | 43.5 | 59.2 | 49.7 | 87.3 | 88.1 | 51.8 | 46.5 | 49.1 | 41.3 | **44.5** | 42.9 |
| +ViCrop | 47.8 | 57.9 | 51.8 | **88.0** | **88.7** | **66.3** | 40.3 | **53.3** | 44.5 | 37.0 | 40.6 |
| +SD-RPN(Ours) | **60.9** | **65.8** | **62.8**$_{↑12.0}$ | 87.9$_{↑2.0}$ | 88.6$_{↑1.5}$ | 58.8 | **47.0** | 52.9$_{↑9.9}$ | **51.8** | 39.8 | **45.8**$_{↑9.2}$ |
| **DeepSeek-VL-1.3B** | 34.8 | 55.3 | 42.9 | 86.1 | 87.1 | 40.5 | 33.0 | 36.8 | 30.1 | 29.3 | 30.1 |
| +SD-RPN(Ours) | **55.7** | **59.2** | **57.1**$_{↑14.2}$ | **87.5**$_{↑1.4}$ | **88.3**$_{↑1.2}$ | **56.8** | **34.0** | **45.4**$_{↑8.6}$ | **44.8** | **32.5** | **38.6**$_{↑8.5}$ |
| **DeepSeek-VL-7B** | 37.4 | 50.0 | 42.4 | 86.0 | 87.1 | 47.5 | **41.5** | 44.5 | 37.3 | **41.3** | 39.3 |
| +SD-RPN(Ours) | **52.2** | **55.3** | **53.4**$_{↑11.0}$ | **88.2**$_{↑2.2}$ | **89.0**$_{↑1.9}$ | **58.8** | 39.8 | **49.3**$_{↑4.8}$ | **52.3** | 40.0 | **46.1**$_{↑6.8}$ |
| **Qwen2.5-VL-3B** | 81.7 | 60.5 | 73.3 | 87.8 | 88.6 | 80.5 | 52.0 | 66.3 | 70.0 | 47.8 | 58.9 |
| +SD-RPN(Ours) | **91.3** | **65.8** | **81.2**$_{↑7.9}$ | **88.9**$_{↑1.1}$ | **89.3**$_{↑0.7}$ | **88.0** | **58.3** | **73.1**$_{↑6.8}$ | **79.5** | **51.0** | **65.3**$_{↑6.4}$ |
| **Qwen2.5-VL-7B** | 80.0 | 75.0 | 78.0 | 86.7 | 87.9 | 81.8 | 62.8 | 72.5 | 72.8 | 54.5 | 63.6 |
| +Thyme† | 83.5 | 80.3 | 82.2 | - | - | 91.0 | 63.0 | 77.0 | 86.5 | 57.5 | 72.0 |
| +DeepEyes† | 91.3 | **88.2** | **90.1** | 87.7 | - | 91.3 | 59.0 | 75.1 | **86.6** | 58.5 | 72.6 |
| +SD-RPN(Ours) | **96.5** | 79.0 | 89.5$_{↑11.5}$ | **88.4**$_{↑1.7}$ | **88.9**$_{↑1.0}$ | **92.5** | **64.5** | **78.5**$_{↑6.0}$ | 86.5 | **60.5** | **73.5**$_{↑9.9}$ |

where ∘ represents a masking operation that uses $\mathcal{B}$ to select the foreground features from the encoder's output. Box Upscaling can achieve a higher effective resolution for small, individual regions and Masked Upscaling better preserves the global spatial and positional relationships between all foreground elements, which is crucial for structured data. In the second stage, the high-resolution visual tokens are inserted into the sequence immediately following the original visual tokens. The LLM then performs auto-regressive decoding on this augmented context to generate the final answer. We present a empirical comparison of these two upscaling strategies in our ablation study.

## 4 EXPERIMENTS

### 4.1 EXPERIMENT CONFIGURATIONS.

**Benchmark Settings.** To ensure a comprehensive evaluation, we test our SD-RPN framework across a range of benchmarks, following the protocols of established works (Shi et al., 2025a; Zhang et al., 2025a; Zheng et al., 2025). Our results are presented in two parts to clearly delineate perfor-

mance across different domains. The first part focuses on the Document & OCR category, which assesses performance on text-rich images, and includes five benchmarks: DocVQA (Mathew et al., 2020), ChartQA (Masry et al., 2022), OCRBench (Liu et al., 2024b), InfoVQA (Mathew et al., 2022), and TextVQA (Singh et al., 2019). The second part focuses on Vision-Centric and High-Resolution tasks, including V-Star Bench (V*) (Wu & Xie, 2023), POPE (Li et al., 2023), and HR-Bench (Wang et al., 2024b). Notably, for TextVQA, we do not provide the model with external OCR annotations, a strategy adopted from ViCrop (Zhang et al., 2025a) to ensure a fair evaluation of the MLLM's intrinsic perceptual capabilities.

**Baselines and Comparison Methods.** To evaluate SD-RPN, we integrate it into three prominent MLLM families. Our primary experiments are conducted on the widely-used LLaVA-1.5 (Liu et al., 2023a) architecture, across both its 7B and 13B scales. To demonstrate the generalizability of our approach, we also apply it to the more recent DeepSeek-VL (Lu et al., 2024) and Qwen2.5-VL (Bai et al., 2025) models. We compare our method against several representative techniques for high-resolution visual processing. For the LLaVA-1.5 models, we benchmark against $S^2$ (Shi et al., 2024), a full-tuning method, and ViCrop (Zhang et al., 2025a), a training-free cropping baseline. For the Qwen2.5-VL model, we additionally compare against DeepEyes (Zheng et al., 2025) and Thyme (Zhang et al., 2025b), which are recent reinforcement learning-based method. To ensure a rigorous and fair comparison for full reproducibility, we have re-implemented all competing methods within the unified lmms-eval evaluation library (Zhang et al., 2024a), unless noted otherwise.

**Implementation Details.** To train our proposed SD-RPN, we generate pseudo-labels from a combined dataset of GQA (Hudson & Manning, 2019) and OCR-VQA (Mishra et al., 2019), which provides supervision for natural scenes and text-rich images, respectively. These pseudo-labels are derived from the models' internal response-to-image attention maps. Inspired by recent analyses of MLLM attention mechanisms (Kang et al., 2025b), we select attention maps from the middle layers for this process. During the pseudo-label generation pipeline, the relative attention thresholds for defining the foreground ($\mathcal{S}_{fg}$) and background ($\mathcal{S}_{bg}$) sets are empirically set to 0.2 and 0.1, respectively. For experiments involving Qwen2.5-VL, which natively supports dynamic resolutions, we standardize the input for consistency. On most benchmarks, we set the number of visual tokens to 576 to align with the LLaVA and DeepSeek-VL baselines. For high-resolution benchmarks, we increase this to a maximum of 4096 visual tokens to effectively process the detailed inputs. Further details on the layer selection rationale and hyper-parameters can be found in Appendix A.

## 4.2 MAIN RESULTS

We present our main results in Tab. 1 for text-rich benchmarks and Tab. 2 for vision-centric benchmarks. Our evaluation assesses both performance and efficiency.

**Performance.** Integrating our SD-RPN framework yields substantial and consistent performance gains across all evaluated MLLM families. On the text-rich benchmarks (Tab. 1), our method boosts the average score by approximately 7% over the LLaVA-1.5 and DeepSeek-VL baselines and 3% over the more powerful Qwen2.5-VL baseline. The benefits are also evident on vision-centriand high-resolution tasks (Tab. 2), where our approach achieves average improvements of over 10% on V* Bench and over 6% on HR-Bench. Notably, on high-resolution benchmarks, SD-RPN achieves performance comparable or superior to DeepEyes which relies visual Chain-of-Thought (CoT) that incurs substantially greater computational cost. To ensure a fair comparison on V-Star Bench, we note that the image cropping strategy employed by ViCrop was disabled during our evaluation. Beyond these quantitative results, Fig. 5 offers a qualitative comparison with the LLaVA-1.5 and Qwen2.5-VL baselines. Additional visual examples are provided in Appendix E.

**Efficiency.** The two-stage inference process used by our method and ViCrop inherently involves a trade-off, reducing throughput in exchange for higher accuracy. This additional latency stems from three primary sources: the initial RoI prediction stage, the feature extraction for the high-resolution RoI crops, and the increased number of visual tokens processed by the LLM in the final generation stage. However, our SD-RPN is architected for superior efficiency in the critical RoI prediction step. By leveraging only a subset of the MLLM's layers in a single forward pass, our method is significantly faster than competing two-stage approaches. For example, when integrated into the LLaVA-1.5-7B model, our RoI prediction stage is 1.5× faster than that of ViCrop, demonstrating a more favorable balance between performance and computational cost.

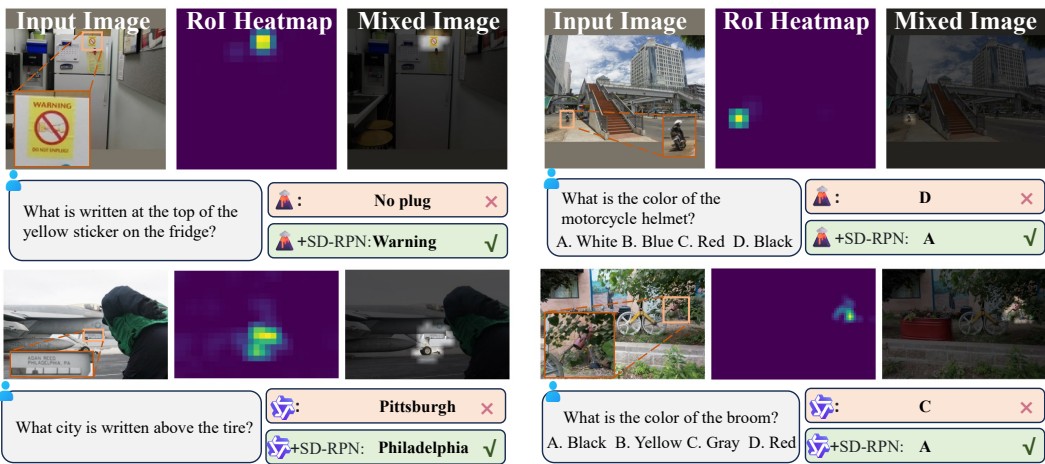

Figure 5: **Qualitative comparisons on TextVQA and V-Star benchmarks.** These examples show challenging cases where the baseline model fails while our SD-RPN succeeds. For each question, our model generates a dense **RoI Heatmap** (center) to localize the relevant area. The **Mixed Image** (right) shows the final high-confidence region identified by SD-RPN, which is used for fine-grained perception to find the correct answer.

## 4.3 ABLATION STUDY

Table 3: Ablation study on the key components of our method.

| (#) | Setting | Throughput | OCRBench | TextVQA | POPE | V* | Ave. |
|---|---|---|---|---|---|---|---|
| (0) | LLaVA-1.5-7B (Baseline) | 1.0× | 31.4 | 46.1 | 85.9 | 50.3 | 53.4 |
| (1) | Response-to-Image Attention | 0.42× | 32.8$_{\uparrow2.4}$ | 53.0$_{\uparrow6.9}$ | 84.3$_{\downarrow1.6}$ | 57.6$_{\uparrow7.6}$ | 56.9$_{\uparrow3.8}$ |
| (2) | Attention Prediction | 0.58× | 32.2$_{\uparrow0.8}$ | 56.8$_{\uparrow10.7}$ | 85.7$_{\downarrow0.2}$ | 61.3$_{\uparrow10.0}$ | 59.0$_{\uparrow5.3}$ |
| (3) | +Label Assignment | 0.55× | 33.9$_{\uparrow2.5}$ | 57.3$_{\uparrow11.2}$ | 85.6$_{\downarrow0.3}$ | 68.6$_{\uparrow18.3}$ | 61.4$_{\uparrow7.9}$ |
| (4) | +Remove Sink Tokens | 0.55× | 36.2$_{\uparrow4.8}$ | 57.9$_{\uparrow11.8}$ | 86.8$_{\uparrow0.9}$ | **68.6**$_{\uparrow18.3}$ | 62.4$_{\uparrow9.0}$ |
| (5) | +Masked Upscaling | **0.62×** | **37.0**$_{\uparrow5.6}$ | **58.7**$_{\uparrow12.6}$ | **87.1**$_{\uparrow1.2}$ | 67.5$_{\uparrow17.2}$ | **62.6**$_{\uparrow9.2}$ |

In this section, we first present ablations for our core design choices, including the pseudo-label generation process, backbone layer depth, and data efficiency. These experiments were conducted using default thresholds ($\tau_{fg} = 0.2, \tau_{bg} = 0.1$). We then provide a detailed ablation study on these $\tau_{fg}$ and $\tau_{bg}$ thresholds to determine their optimal values.[1]

**On Pseudo-Label Gerneration & SD-RPN.** Tab. 3 presents a detailed ablation study dissecting the core components of our proposed framework. We begin by evaluating two baseline strategies for leveraging attention. Box upscaling is adopted as the upscaling setting except additional note. In (#1), we use the raw Response-to-Image Attention map directly for RoI identification. In (#2), our RPN is trained to regress these attention scores via a Mean Squared Error (MSE) loss. While both methods improve upon the baseline, their average gains of 3.8% and 5.3% are limited, confirming our hypothesis that using noisy, unprocessed attention maps for direct supervision is suboptimal. The introduction of Label Assignment strategy (#3), which creates high-confidence labels and ignores ambiguous regions, yields a significant performance jump to a 7.9% average improvement. This is further enhanced by Remove Sink Tokens step (#4), which denoises the attention map to achieve a 9.0% average gain. These results validate that our proposed denoising pipeline is crucial for generating high-quality supervision and enabling precise RoI prediction. Aligning with previous analysis in Sec. 3.4, masked upscaling achieves superior performance on OCRBench and TextVQA. Given its better throughput (0.62x vs. 0.55x), we adopt it as the default setting.

**On Backbone Layers.** In Tab. 4a, we conduct an ablation study on the number of frozen backbone layers (B) that the RPN is built upon, keeping the number of trainable RPN layers fixed. We observe a clear trend in performance: as the backbone depth increases, the average performance steadily improves, reaching its peak with the B15R3 configuration, which achieves a 9.2% gain over

---

[1]We thank the reviewers for motivating this deeper analysis of our hyperparameter settings.

Table 4: **(a)** Varying the number of frozen backbone layers (B) while keeping the RPN layers fixed. **(b)** Varying the number of training data samples. The † symbol indicates that pseudo-labels were generated using ground-truth responses from the LLaVA supervised fine-tuning dataset. All scores are accuracy (%) except for POPE, which reports the F1-score.

| Setting | Through-put | OCR-Bench | Text-VQA | POPE | V* | Ave. |
|---|---|---|---|---|---|---|
| Baseline | $1.0\times$ | 31.4 | 46.1 | 85.9 | 50.3 | 53.4 |
| B3R3 | $\mathbf{0.71}\times$ | $33.9_{\uparrow 2.5}$ | $51.0_{\uparrow 4.9}$ | $87.0_{\uparrow 1.1}$ | $53.9_{\uparrow 3.6}$ | $56.5_{\uparrow 3.1}$ |
| B6R3 | $0.63\times$ | $34.4_{\uparrow 3.0}$ | $53.5_{\uparrow 7.4}$ | $\mathbf{87.7}_{\uparrow 1.8}$ | $57.1_{\uparrow 6.8}$ | $58.2_{\uparrow 4.8}$ |
| B9R3 | $0.66\times$ | $35.8_{\uparrow 4.4}$ | $56.3_{\uparrow 10.2}$ | $87.1_{\uparrow 1.2}$ | $59.7_{\uparrow 9.4}$ | $59.7_{\uparrow 6.3}$ |
| B12R3 | $0.60\times$ | $35.5_{\uparrow 4.1}$ | $57.8_{\uparrow 11.7}$ | $87.4_{\uparrow 1.5}$ | $67.0_{\uparrow 16.7}$ | $61.9_{\uparrow 8.5}$ |
| B15R3 | $0.62\times$ | $\mathbf{37.0}_{\uparrow 5.6}$ | $\mathbf{58.7}_{\uparrow 12.6}$ | $87.1_{\uparrow 1.2}$ | $\mathbf{67.5}_{\uparrow 17.2}$ | $\mathbf{62.6}_{\uparrow 9.2}$ |
| B18R3 | $0.52\times$ | $35.5_{\uparrow 4.1}$ | $57.8_{\uparrow 11.7}$ | $87.3_{\uparrow 1.4}$ | $66.0_{\uparrow 15.7}$ | $61.7_{\uparrow 8.3}$ |

(a)

| Setting | OCR-Bench | Text-VQA | POPE | V* | Ave. |
|---|---|---|---|---|---|
| Baseline | 31.4 | 46.1 | 85.9 | 50.3 | 53.4 |
| 10K | $35.2_{\uparrow 3.8}$ | $58.3_{\uparrow 12.2}$ | $\mathbf{87.6}_{\uparrow 1.7}$ | $61.3_{\uparrow 11.0}$ | $60.6_{\uparrow 7.2}$ |
| 25K | $35.8_{\uparrow 4.4}$ | $58.5_{\uparrow 12.4}$ | $87.1_{\uparrow 1.2}$ | $59.7_{\uparrow 9.4}$ | $60.3_{\uparrow 6.9}$ |
| 50K | $\mathbf{37.5}_{\uparrow 6.1}$ | $58.3_{\uparrow 12.2}$ | $86.9_{\uparrow 1.0}$ | $67.0_{\uparrow 16.7}$ | $62.4_{\uparrow 9.0}$ |
| 100K | $36.5_{\uparrow 5.1}$ | $58.3_{\uparrow 12.2}$ | $86.8_{\uparrow 0.9}$ | $66.0_{\uparrow 15.7}$ | $61.9_{\uparrow 8.5}$ |
| 152K | $37.0_{\uparrow 5.6}$ | $\mathbf{58.7}_{\uparrow 12.6}$ | $87.1_{\uparrow 1.2}$ | $\mathbf{67.5}_{\uparrow 17.2}$ | $\mathbf{62.6}_{\uparrow 9.2}$ |
| 152K† | $35.2_{\uparrow 3.8}$ | $56.9_{\uparrow 10.8}$ | $86.6_{\uparrow 0.7}$ | $63.9_{\uparrow 13.6}$ | $60.7_{\uparrow 7.3}$ |

(b)

the baseline. Beyond this point, performance begins to decline. The relationship between backbone depth and efficiency is more nuanced. While the inference cost of the RoI prediction stage is positively correlated with the number of backbone layers, the overall throughput is not monotonic. This is because a more precise RoI prediction, often produced by a deeper backbone, can reduce the number of irrelevant or noisy visual tokens that are passed to the second, more costly generation stage. This interplay between the increasing cost of the first stage and the potentially decreasing cost of the second stage explains the fluctuating throughput values.

**On Data Efficency.** We conduct ablation studies on the size of the training set, with results presented in Tab. 4b. Performance generally improves with an increased number of training samples. Remarkably, even when trained on only 10K samples, the framework achieves substantial gains over the baseline (e.g., over 10% on improvement on both TextVQA and V-Star). Further increases in data size continue to yield performance gains. The peak performance is ultimately achieved when training on the full combined dataset of 152K samples. The final row of Tab. 4b presents a crucial comparison where we generate pseudo-labels using ground-truth responses from the LLaVA supervised fine-tuning (SFT) dataset. Counter-intuitively, this approach yields a notably lower average performance of 60.7% compared to the 62.6% achieved by our standard self-distillation method. This suggests that the attention maps produced by the model's own generated responses provide a more effective and internally consistent supervision signal for distilling localization knowledge.

Table 5: Ablation study on pseudo-label thresholds $\tau_{fg}$ and $\tau_{bg}$.

| $\tau_{fg}$ | $\tau_{bg}$ | OCR-Bench | Text-VQA | POPE | V* | Ave. |
|---|---|---|---|---|---|---|
| Baseline | | 31.4 | 46.1 | 85.9 | 50.3 | 53.4 |
| 0.10 | 0.10 | 35.1 | 57.4 | 87.4 | 65.4 | 61.3 |
| 0.15 | 0.10 | 36.3 | 57.9 | 87.2 | 67.5 | 62.2 |
| 0.20 | 0.10 | 37.0 | 58.7 | 87.1 | 67.5 | 62.6 |
| 0.25 | 0.10 | 37.0 | 58.6 | 86.7 | 68.1 | 62.6 |
| 0.30 | 0.10 | 36.6 | 58.1 | 86.9 | 64.9 | 61.6 |
| 0.20 | 0.075 | 36.7 | 58.4 | 86.9 | **70.7** | 63.2 |
| **0.20** | **0.05** | 37.3 | **58.8** | 87.2 | **70.7** | **63.5** |
| 0.20 | 0.03 | **37.8** | 58.7 | **87.6** | 68.6 | 63.2 |

**On Foreground and Background Threshold.** Tab. 5 ablates the pseudo-label thresholds. We find that setting $\tau_{fg} = \tau_{bg} = 0.10$ (a simple binary classification) performs poorly (61.3% avg.), as it suffers from ambiguous tokens. By optimizing the thresholds to $\tau_{fg} = 0.20$ and $\tau_{bg} = 0.05$, we establish a new, stronger default. This setting achieves a 63.5% average performance, surpassing our original default (62.6%) and the baseline (53.4%) by 0.9% and 10.1%, respectively.

## 5 CONCLUSION

We proposed SD-RPN, a self-distilled region proposal framework that efficiently exploits the intrinsic localization signals of MLLMs to identify Regions of Interest without external annotations or auto-regressive decoding. By attaching a RPN to frozen backbones and training it with denoised pseudo-labels, our method achieves consistent improvements in fine-grained perception while maintaining strong efficiency and generalization. Extensive experiments confirm its advantage over both full-image scaling and training-free heuristics, and ablations highlight its robustness and data efficiency. Our work establishes a principled direction for scalable high-resolution perception in MLLMs, and opens avenues toward adaptive token allocation and broader multimodal applications such as video and document understanding.

## 6 ACKNOWLEDGEMENTS

This work was supported in part by the Start-up Grant (No. 9610680) of the City University of Hong Kong, Young Scientist Fund (No. 62406265) of NSFC, and the Australian Research Council under Projects DP240101848 and FT230100549.

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

## A  MORE IMPLEMENTATION DETAILS.

**Training Setting.** The hyperparameters used to train our RPN were adapted from TwigVLM (Shao et al., 2025) and are detailed in Table 6. All models were trained on a server using four NVIDIA A6000 GPUs. Our training dataset consists of 152K samples, with 72K sourced from the GQA dataset (primarily natural images) and 80K from the OCR-VQA dataset (text-rich images). As a point of reference for computational cost, training the RPN for the LLaVA-1.5-7B model on our full 152K-sample dataset completes in around two hours.

Table 6: Training hyperparameters for the RPN.

| Config | Setting |
|---|---|
| optimizer | AdamW |
| weight decay | 0. |
| optimizer momentum | $\beta_1, \beta_2 = 0.9, 0.98$ |
| batch size | 128 |
| learning rate schedule | cosine decay |
| peak learning rate | 5e-5 |
| warm-up strategy | linear |
| warm-up ratio | 0.03 |
| training epochs | 1 |

Table 7: Model-specific layer configurations. "Selected Attention Layers" refers to the layers from which attention maps were extracted for pseudo-label generation. "Frozen Backbone Layers (B)" refers to the number of initial layers kept frozen, upon which the trainable RPN is built.

| Model | Selected Attention Layers | Frozen Backbone Layers (B) |
|---|---|---|
| LLaVA-1.5-7B | 14 | 15 |
| LLaVA-1.5-13B | 13–16 | 15 |
| DeepSeek-VL-1.3B | 7–14 | 9 |
| DeepSeek-VL-7B | 9–17 | 12 |
| Qwen2.5-VL-3B | 20–22 | 21 |
| Qwen2.5-VL-7B | 16–23 | 16 |

**Layer Selection Rationale.** The specific layers used for pseudo-label generation and for defining the RPN backbone were empirically chosen for each model architecture, with the final configurations detailed in Table 7. For generating pseudo-labels, our selection of attention maps is inspired by recent analyses (Kang et al., 2025b) which demonstrate that the middle layers of an MLLM contain the most potent visual grounding signals. These intermediate layers represent a "sweet spot" where visual and textual representations are sufficiently fused for localization, yet before the model's final layers become overly specialized for abstract reasoning and text generation, potentially weakening the direct visual signal. When a range of layers is specified, the attention maps were averaged to create a more stable and robust signal. The number of frozen backbone layers, B, determines the depth and richness of the features that the trainable RPN operates on. This choice represents a critical trade-off. A deeper backbone (larger B) provides more semantically rich features but increases the computational cost of the RoI prediction stage and can limit the learning capacity of a small RPN. Conversely, a shallower backbone is faster but may not provide features that are sufficiently aligned for accurate RoI prediction. The values in Table 7 were determined through ablation studies (see Table 4a) to find the optimal balance between performance and efficiency for each model.

**Algorithm Details.** To provide a concrete and reproducible specification of our training procedure, we detail the complete end-to-end process in Algorithm 1. The algorithm consists of a main training loop and the core helper routines. The main training loop iterates through the dataset, performing two key stages for each sample. First, the "teacher" stage (lines 3-10) uses the full, frozen MLLM to generate a pseudo-label. This involves auto-regressively generating a response, extracting the raw response-to-image attention map from a pre-selected middle layer, and then refining this map using our RemoveSinkTokens and AssignLabels helper functions. This produces the final denoised

---

**Algorithm 1** Self-Distilled RPN: training and helper routines

---

**Require:** Full MLLM $\mathcal{L}$ with $L$ layers; frozen depth $B$; RPN depth $R$; pseudo-label layer $l$; dataset $\mathcal{D}$; thresholds $(\tau_{\mathrm{norm}}, \tau_{fg}, \tau_{bg})$; optimizer for trainable RPN parameters

**Ensure:** Trained RPN on top of the frozen backbone

1: **Initialization:** Initialize RPN layers $B{+}1{:}B{+}R$ from $\mathcal{L}$; freeze layers $1{:}B$; initialize optimizer.
2: **for** each $(x_v, x_t) \in \mathcal{D}$ **do**
             ▷ // Teacher: generate pseudo-labels from the full MLLM
3:     $Y_{\mathrm{pred}} \leftarrow Decode(\mathcal{L}, x_v, x_t)$                      ▷ Auto-regressive response
4:     $x_t^{\mathrm{full}} \leftarrow Concat(x_t, Y_{\mathrm{pred}})$
5:     $\mathbf{H}_v^l, \mathbf{Q}_r^l, \mathbf{K}_v^l \leftarrow Features(\mathcal{L}, x_v, x_t^{\mathrm{full}}, l)$
6:     $\mathbf{A} \leftarrow Softmax\left(\mathbf{Q}_r^l (\mathbf{K}_v^l)^\top / \sqrt{d}\right)$             ▷ Softmax over visual tokens
7:     $\mathbf{M}_{\mathrm{RoI}} \leftarrow MeanAcrossHeadsAndResp(\mathbf{A})$
8:     $\mathbf{M}_{\mathrm{RoI}}' \leftarrow RemoveSinkTokens(\mathbf{M}_{\mathrm{RoI}}, \mathbf{H}_v^l, \tau_{\mathrm{norm}})$
9:     $\bar{\mathbf{M}}_{\mathrm{RoI}} \leftarrow AssignLabels(\mathbf{M}_{\mathrm{RoI}}', \tau_{fg}, \tau_{bg})$           ▷ Values in $\{-1, 0, 1\}$
10:    $mask \leftarrow (\bar{\mathbf{M}}_{\mathrm{RoI}} \neq -1)$
           ▷ // Student: forward through frozen $1{:}B$ and trainable $B{+}1{:}B{+}R$
11:    $\mathbf{H}_v, \mathbf{H}_{\mathrm{RoI}} \leftarrow Features(\mathcal{L}_{\mathrm{RPN}}, x_v, x_t, B{+}R{-}1)$
12:    $\mathbf{Q}_{\mathrm{RoI}} \leftarrow LP_q(Norm(\mathbf{H}_{\mathrm{RoI}})), \quad \mathbf{K}_v \leftarrow LP_k(Norm(\mathbf{H}_v))$
13:    $\hat{\mathbf{M}}_{\mathrm{RoI}} \leftarrow \mathbf{Q}_{\mathrm{RoI}} \mathbf{K}_v^\top$                      ▷ Logits for dense RoI map
14:    $loss \leftarrow BCEWithLogits\left(\hat{\mathbf{M}}_{\mathrm{RoI}}[mask], \bar{\mathbf{M}}_{\mathrm{RoI}}[mask]\right)$
15:    Update(optimizer, $loss$)

**Helper routines**

1: **function** REMOVESINKTOKENS($\mathbf{M}, \mathbf{H}_v, \tau_{\mathrm{norm}}$)
2:     **return** $\mathbf{M}$ with entries set to 0 where $\|\mathbf{H}_{v,j}\|_2 > \tau_{\mathrm{norm}}$

3: **function** ASSIGNLABELS($\mathbf{M}, \tau_{fg}, \tau_{bg}$)
4:     $a_{\max} \leftarrow \max_j \mathbf{M}_j$
5:     $\mathcal{S}_{fg} \leftarrow \{j : \mathbf{M}_j \geq \tau_{fg}\, a_{\max}\}$
6:     $\mathcal{B}_{fg} \leftarrow MinEnclosingBox(\mathcal{S}_{fg})$
7:     $\mathcal{S}_{bg} \leftarrow \{j \notin \mathcal{B}_{fg} : \mathbf{M}_j \leq \tau_{bg}\, a_{\max}\}$
8:     Build $\bar{\mathbf{M}}$ with $\bar{\mathbf{M}}_j{=}1$ for $j{\in}\mathcal{S}_{fg}$, $\bar{\mathbf{M}}_j{=}0$ for $j{\in}\mathcal{S}_{bg}$, and $-1$ otherwise
9:     **return** $\bar{\mathbf{M}}$

---

Table 8: Ablation on the region proposal layers.

| Setting | OCRBench | TextVQA | POPE | V* | Ave. |
|---|---|---|---|---|---|
| LLaVA-1.5-7B | 31.4 | 46.1 | 85.9 | 50.3 | 53.4 |
| B15R1 | 35.8↑4.4 | 54.7↑8.6 | 87.1↑1.2 | 56.0↑5.7 | 58.4↑5.0 |
| B15R2 | 35.9↑4.5 | 57.8↑11.7 | **87.2**↑1.3 | 63.4↑13.1 | 61.1↑7.7 |
| B15R3 | 37.0↑5.6 | **58.7**↑12.6 | 87.1 ↑1.2 | **67.5**↑17.2 | **62.6**↑9.2 |
| B15R4 | **37.1**↑5.7 | 58.4↑12.3 | 87.1↑1.2 | **67.5**↑17.2 | 62.5↑9.1 |

pseudo-label, $\bar{\mathbf{M}}_{\mathrm{RoI}}$. Second, the "student" stage (lines 11-15) trains the RPN. The RPN performs a forward pass on the original input to predict a dense RoI map $\hat{\mathbf{M}}_{\mathrm{RoI}}$. A masked BCEWithLogits loss is then computed between the predicted map and the pseudo-label target, ensuring that ambiguous regions (where the mask is false) do not contribute to the gradient. Finally, the RPN's parameters are updated via backpropagation.

## B    MORE ABLATIONS

These experiments blow were conducted using default thresholds ($\tau_{fg} = 0.2$, $\tau_{bg} = 0.1$), as they were implemented before threshold ablation.

Table 9: Ablation on the setting of Pre-smoothing and Post-smoothing.

| Setting | OCRBench | TextVQA | POPE | V* | Ave. |
|---|---|---|---|---|---|
| LLaVA-1.5-7B | 31.4 | 46.1 | 85.9 | 50.3 | 53.4 |
| Pre-smoothing | 34.8 $_{\uparrow 3.4}$ | 58.4 $_{\uparrow 8.3}$ | **87.9** $_{\uparrow 2.0}$ | 60.7 $_{\uparrow 10.4}$ | 60.5 $_{\uparrow 7.1}$ |
| Post-smoothing | **37.0** $_{\uparrow 5.6}$ | **58.7** $_{\uparrow 12.6}$ | 87.1 $_{\uparrow 1.2}$ | **67.5** $_{\uparrow 17.2}$ | **62.6** $_{\uparrow 9.2}$ |

Figure 6: **Performance-Throughput Trade-off on the V-Star Benchmark.** Each point on the plot corresponds to a different maximum number of visual tokens. Our approach achieves a superior trade-off. The x-axis is on a logarithmic scale for clarity.

**On Region Proposal Layers**. In Table 8, we present an ablation study on the depth of the trainable Region Proposal Network, varying the number of tunable layers (R). We observe that increasing the RPN's depth from one to three layers yields a significant performance improvement (from 58.4% to 62.6% average score). Beyond this point, we found that the performance gains diminished, failing to justify the additional computational cost incurred during both training and inference. Therefore, to achieve the best balance between model capacity and efficiency, we selected R=3 as the default setting for our framework.

**On Pseudo-label Smoothing**. Our standard pipeline applies a Gaussian filter to the RPN's final predicted map, a step we term post-smoothing. We investigated an alternative approach, pre-smoothing, where the filter is instead applied directly to the pseudo-labels during their generation. As shown in Table 9, the post-smoothing approach consistently outperforms pre-smoothing across most benchmarks, achieving a +2.1% higher average score. We hypothesize that this performance gap stems from a distributional shift induced by pre-smoothing. Applying a filter to the sparse pseudo-labels makes them denser, effectively enlarging the target foreground area. While this might appear beneficial, it creates a significant discrepancy between the dense, artificially softened labels and the sharper, more sparse predictions natural to the self-initilized RPN. This distribution gap complicates the training process, particularly in our data-efficient setting. This difficulty is reflected in the final convergence loss, where the pre-smoothing strategy results in a 2.5x higher value than post-smoothing (0.05 vs. 0.02). We conclude that forcing the network to learn these artificially blurred targets increases ambiguity, ultimately degrading the precision of the RoI predictions.

## C  MORE COMPARISON AND ANALYSIS

### C.1  ANALYSIS OF THE PERFORMANCE-THROUGHPUT TRADE-OFF

More recent MLLMs, such as Qwen2.5-VL, adopt dynamic resolution techniques that handle images of various sizes by producing a varying number of visual tokens. A key concern is whether our RoI-selection approach is more effective than simply increasing the native input resolution of the baseline model. To address this, we analyze the trade-off between performance and efficiency by scaling the input resolution for both the Qwen2.5-VL baseline and our proposed SD-RPN. The results of this analysis on the V-Star benchmark are presented in Fig. 6. The figure plots performance against throughput (tokens/s), where the ideal model would be in the top-right corner. It is evident that the trade-off curve for our method is consistently superior to that of the baseline. For example, on Qwen2.5-VL-7B, our method achieves a performance of 77.5 at 5.1 tokens/s using a 576-resolution input, a level the baseline only reaches at 4096 resolution with a much lower throughput of 1.0

tokens/s. This demonstrates that SD-RPN provides a significantly better balance between accuracy and computational cost, confirming the effectiveness of our approach over naive resolution scaling.

## C.2 COMPARISON TO EXTERNAL GROUNDING MODELS

A practical concern regarding our SD-RPN is its computational cost, as it utilizes a portion of the MLLM. It may seem counter-intuitive, but our SD-RPN is, in fact, significantly **more computationally efficient** than using an external grounding model like Grounding-DINO or YOLO-World. The reason is that our RPN **re-uses computations** that the MLLM must perform anyway. An external model (e.g., Grounding-DINO) would require **redundant computation** in the form of a *separate, full* image encoding pass, which is computationally expensive. In contrast, **our approach** uses **feature re-use**: our RPN attaches to the MLLM's middle layers and operates on features that are *already computed* during the MLLM's initial forward pass. The *only* extra cost is passing these features through the RPN layers (e.g., 3 layers in our B15R3 configuration), which is a minimal addition.

To provide a concrete analysis, we benchmarked our method (on LLaVA-1.5-7B) against a Grounding-DINO-Base model on a NVIDIA A6000 GPU. The results, presented in Table 10, are clear. Our RPN is **over 20x faster** in its specific prediction step (10.3ms vs 231.2ms) because it avoids a redundant, full-image encoding. This also leads to a much faster overall evaluation time on the V* benchmark. Our method is also **significantly more accurate** (62.6% vs 57.6% avg.). This is because our RPN is an integrated part of the MLLM and has access to its *contextual understanding* of the query. External models are excellent at finding *literal objects* (e.g., "the bottom man") but fail at abstract, reasoning-based queries that don't name a specific object, such as the example in the last row of Figure 7. Our RPN, trained on the MLLM's internal attention, excels at this.

Table 10: Efficiency and Performance: Ours (SD-RPN) vs. External Grounding-DINO. The "Grounding-DINO-B (1Box)" method crops one sub-image using the single most confident bounding box, while "(2Box)" crops two sub-images using the top 2 boxes.

| Method | RPN Time (ms) | V* Eval Time (s) | OCRBench | TextVQA | POPE | V* | Ave. |
|---|---|---|---|---|---|---|---|
| **Ours (SD-RPN)** | **10.3** | **73** | **37.0** | **58.7** | **87.1** | **67.5** | **62.6** |
| Grounding-DINO-B (2Box) | 231.2 | 248 | 32.5 | 54.5 | 82.4 | 62.3 | 57.6 |
| Grounding-DINO-B (1Box) | 231.2 | 113 | 32.5 | 52.6 | 86.9 | 56.5 | 57.1 |

## C.3 DEEPER ANALYSIS ON SOURCES OF ATTENTION NOISE

In our work, we identify and actively mitigate two primary sources of noise from the raw attention maps used to generate pseudo-labels. First, there are **Type 1: Sink Tokens**. We build on recent studies (Darcet et al., 2023; Kang et al., 2025a) which identify "sink tokens"—visual tokens that attract high attention despite being semantically irrelevant to the grounded object. Second, there are **Type 2: Ambiguous Foreground/Background Signals**. This is a more common, semantic, and task-dependent form of noise. As illustrated in Fig. 3 and described in Section 3.2, these are "noisy maps" that exhibit "erroneously high attention in background areas and incomplete activation across the true foreground object."

As the first type of sink token belongs to a well-defined source of noise, we now investigate the second type. Our empirical study finds that questions or images in the following three categories are prone to producing noisy attention maps, as illustrated in Fig. 7. The first source of noise is **cluttered scenes or dense text** (e.g., the first sample in Fig. 7); in these cases, the model's attention can be fraught with noise as it diffuses across multiple irrelevant-but-plausible regions. The second type of noise is caused by **abstract or reasoning-heavy queries**. In practice, we find that a simple query (e.g., "find the cat") tends to produce clean attention. However, many VQA queries are abstract, such as the second sample in Fig. 7. For such queries, the attention map reflects the model's complex *reasoning process*, which can appear noisy as it attends to multiple pieces of related (but not all relevant) information before settling on an answer. The last type of noise is related to **incomplete activation**, where the attention map only weakly or partially covers the true region of interest, as shown in the last sample of Fig. 7.

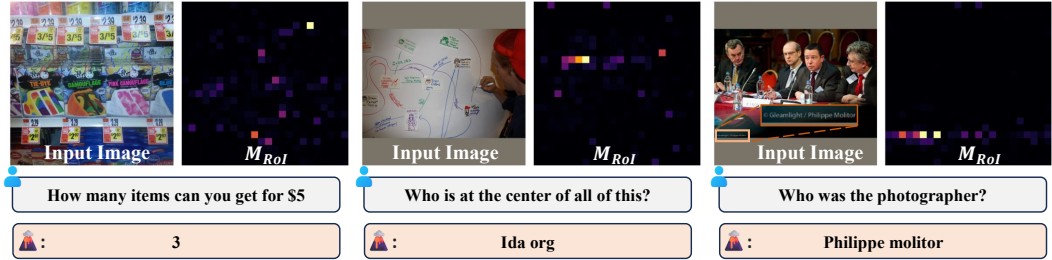

Figure 7: Examples of noisy and sparse attention maps for abstract queries.

## C.4 Comparison with Token Compression Methods

We provide a detailed comparison of our SD-RPN with other SoTA methods designed for high-resolution visual understanding in this section. To facilitate a fair and transparent comparison, Table 11 summarizes the performance of our method against several competitive models on key document and OCR benchmarks. The table also highlights critical architectural and data differences, such as the base Large Language Model (LLM), the size of the training dataset, and the maximum input resolution, all of which are significant factors that influence final performance.

Table 11: Comparison with SoTA high-resolution methods on document and OCR benchmarks. #Denotes results obtained using OCR tokens. *Denotes results reported in the LLaVA-TokenPacker-HD paper (Li et al., 2025). The "+152K" in the #Data column indicates the number of additional samples used for fine-tuning our SD-RPN on the corresponding baseline models.

| Method | LLM | #Data | Max Res. | TextVQA$_{val}$ | OCRBench$_{test}$ | DocVQA$_{test}$ | ChartQA$_{test}$ |
|---|---|---|---|---|---|---|---|
| LLaVA-1.5 | Vicuna-7B | 1.2M | 336×336 | 58.2$^{\#}$ | 31.4 | 22.2 | 18.1 |
| LLaVA-1.5 + SD-RPN | Vicuna-7B | +152K | 336×336 | 64.2$^{\#}$ | 37.0 | 35.1 | 20.1 |
| Monkey (Li et al., 2024b) | Qwen-7B | 1.4M | 896×896 | 67.6 | - | 66.5 | 65.1 |
| LLaVA-NeXT (Liu et al., 2024a) | Vicuna-7B | 1.3M | 672×672 | 64.9$^{*}$ | - | - | - |
| Mini-Gemini-HD (Li et al., 2024a) | Vicuna-7B | 2.7M | 536×1536 | 68.4$^{*}$ | 456$^{*}$ | 65.0$^{*}$ | - |
| LLaVA-TokenPacker-HD (Li et al., 2025) | Vicuna-7B | 2.7M | 1088×1088 | 68.0$^{*}$ | 452$^{*}$ | 60.2$^{*}$ | - |
| DeepSeek-VL | Vicuna-7B | - | 1024×1024 | 63.0 | 42.4 | 49.3 | 61.4 |
| DeepSeek-VL + SD-RPN | Vicuna-7B | +152K | 1024×1024 | 71.5 | 47.9 | 64.5 | 63.6 |
| Qwen2.5-VL | Qwen2.5-7B | - | 672×672 | 81.1 | 81.5 | 92.6 | 82.6 |
| Qwen2.5-VL + SD-RPN | Qwen2.5-7B | +152K | 672×672 | 83.5 | 82.9 | 94.4 | 85.5 |

The results in Table 11 lead to two primary conclusions. First, the overall performance of our method, like others, is strongly correlated with the capabilities of the underlying base MLLM. When applied to an earlier baseline such as LLaVA-1.5, our method's absolute scores on benchmarks like DocVQA are understandably lower than SoTA methods built on more advanced foundations, which often leverage richer training sets and higher input resolutions. This performance difference is primarily a reflection of the base model's capacity rather than a limitation of our enhancement module. Second, our SD-RPN module demonstrates consistent efficacy across base models of varying strengths. When integrated with a more competitive baseline like DeepSeek-VL-7B, our method achieves performance on par with or slightly exceeding other leading methods. To further validate this generalizability, we applied SD-RPN to the powerful Qwen2.5-VL model. Even on this SoTA baseline, our method delivers significant and consistent gains across all benchmarks, utilizing an identical and minimal fine-tuning recipe.

This highlights a key advantage of our approach: SD-RPN provides a data-efficient and scalable pathway for enhancing the fine-grained perceptual capabilities of MLLMs. Unlike methods that necessitate extensive supervised fine-tuning on massive datasets, our self-distillation technique offers a resource-efficient solution for improving high-resolution understanding that can readily adapt to the continuous advancements in base MLLMs.

## D    PROMPT USAGE

Below are the prompts used for pseudo-label generation with LLaVA-1.5 and DeepSeek-VL. Each template follows the respective model's standard input format.

**LLaVA-1.5**

```
<image> USER:{question} Answer the question using a single
word or phrase. ASSISTANT:
```

**DeepSeek-VL**

```
<image_placeholder> USER:{question} Answer the question using
a single word or phrase. Assistant:
```

## E    VISUALIZATION

To provide a deeper, qualitative understanding of our model's advantages, Figure 8 presents a series of challenging visual question-answering examples. These samples, drawn from the V-Star, TextVQA, and DocVQA benchmarks, were specifically chosen because they require recognizing and reasoning about fine-grained details within cluttered scenes. The baseline model, which relies on processing the entire image at a global level, often fails to perceive these critical details. In contrast, our SD-RPN first employs its region proposal mechanism to identify the most salient RoI for a given question—visualized by the green bounding boxes for subsequent fine-grained perception. This two-stage approach allows our model to accurately perform tasks like detailed text extraction, object attribute identification, and spatial reasoning, achieving correct answers where the baseline consistently fails.

In addition to the successful cases, we also provide a failure case analysis in Fig. 9. The failure cases can be broadly attributed to three main reasons:

- **Incomplete Activation.** The top-left "cup" case is a representative example. The RPN correctly identifies the measuring cup but only activates on a small portion of it. This partial RoI provides incomplete visual information to the MLLM, leading to an erroneous prediction.

- **Localization Error.** The top-right "book" example illustrates this type. The RPN is supposed to highlight the book's cover to read the name but erroneously highlights an unrelated region. Given this misleading visual cue, the MLLM fails to produce the correct result.

- **Spatial Relationship Error.** The bottom row shows errors of this type. Here, the RPN highlights all relevant objects correctly (e.g., "bowl" and "faucet"). However, the MLLM still produces the wrong result, suggesting that it struggles to preserve and reason about the precise spatial relationships between objects when they are processed as fine-grained crops.

## F    WHY PREDICTING RoI SCORES OUTPERFORMS USING RAW ATTENTION

Let $X$ denote token-level features available to the region-of-interest (RoI) predictor, $Y \in \{0, 1\}$ is the latent foreground (FG) indicator, and $A \in [0, 1]$ is the model's response-to-image attention used as a noisy proxy for $Y$. Define the posterior $\eta(x) := \Pr(Y = 1 \mid X = x)$.

**Regression view (continuous RoI scoring).**    We train a predictor $h$ by minimizing the population squared loss

$$\mathcal{R}(h) \;=\; \mathbb{E}\big[(h(X) - A)^2\big].$$

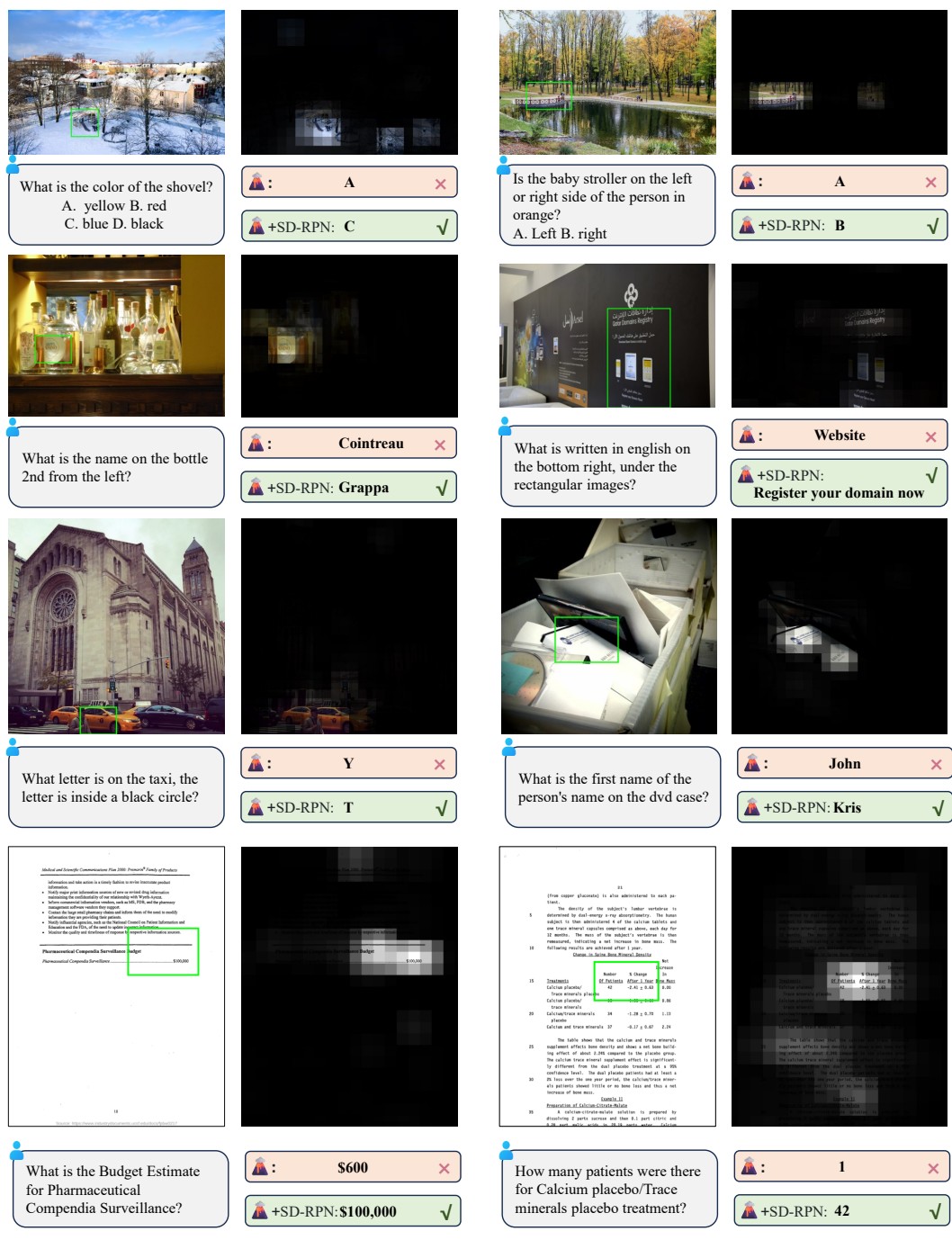

Figure 8: Qualitative comparison of our SD-RPN against the LLaVA-1.5-7B baseline on challenging samples from the V-Star, TextVQA, and DocVQA benchmarks. For each example, the top image displays the original input with the Region-of-Interest (RoI) predicted by our method, while the image below it visualizes the dense RoI map. These examples highlight SD-RPN's ability to precisely localize and analyze fine-grained visual information (e.g., text, small objects), leading to correct answers where the baseline, which processes the full image, consistently fails.

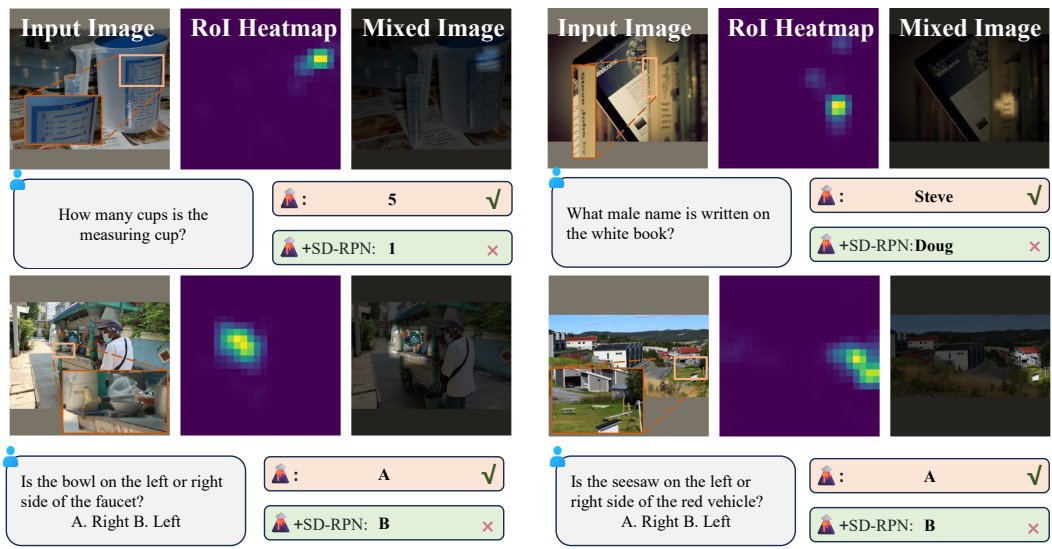

Figure 9: **Failure case analysis of SD-RPN on the LLaVA-1.5-7B baseline.** We present examples where the baseline model succeeds, but our method fails. These failures can be categorized into three main types: **incomplete activation** (top-left), **localization error** (top-right), and **spatial relationship error** (bottom row).

By pointwise conditional minimization, the optimal predictor is the conditional expectation

$$h^\star(x) \;=\; \mathbb{E}[A \mid X = x]. \tag{9}$$

Under standard noise models, $h^\star$ is an affine function of $\eta(x)$:

$$\mathbb{E}[A \mid X = x] = \begin{cases} (1 - \rho_1 - \rho_0)\,\eta(x) + \rho_0, & \text{class-conditional noise (CCN)}, \\ \mu_0 + (\mu_1 - \mu_0)\,\eta(x), & \text{additive activation model}. \end{cases}$$

In both cases, $\mathbb{E}[A \mid X = x]$ is strictly increasing in $\eta(x)$ when $\rho_0 + \rho_1 < 1$ or $\mu_1 > \mu_0$, so thresholding $h^\star$ recovers the Bayes decision boundary up to a constant shift.

**Noise reduction via conditional averaging.** The raw attention signal $A$ can be decomposed into its "signal" component, defined as $h^\star(X) = \mathbb{E}[A \mid X]$, and its "noise" component, $\epsilon = A - \mathbb{E}[A \mid X]$. By construction, the noise is zero-mean conditional on the features ($\mathbb{E}[\epsilon \mid X] = 0$). We can measure the quality of any estimator by its Mean Squared Error (MSE) with respect to this true underlying signal, $h^\star(X)$.

The MSE of the optimal predictor $h^\star(X)$ with respect to the true signal is, trivially, zero:

$$\mathbb{E}\big[(h^\star(X) - h^\star(X))^2\big] = 0.$$

In contrast, the MSE of the raw attention signal $A$ with respect to the true signal is:

$$\mathbb{E}\big[(A - h^\star(X))^2\big] = \mathbb{E}\big[(A - \mathbb{E}[A \mid X])^2\big] = \mathbb{E}\big[\mathbb{E}[(A - \mathbb{E}[A \mid X])^2 \mid X]\big] = \mathbb{E}[\mathrm{Var}(A \mid X)].$$

This term, the expected conditional variance, represents the irreducible error inherent in the attention signal. As long as the attention is not a perfectly deterministic function of the features (i.e., $\mathrm{Var}(A \mid X) > 0$), we have:

$$\mathbb{E}\big[(h^\star(X) - h^\star(X))^2\big] < \mathbb{E}\big[(A - h^\star(X))^2\big].$$

This inequality formally proves that the optimal predictor $h^\star(X)$ is a strictly better, denoised estimate of the underlying signal than the raw attention $A$. Any learned predictor $\hat{h}$ that successfully approximates $h^\star$ will therefore also be a more stable and accurate predictor.

**Classification view (binary RoI selection).** For symmetric class-conditional noise (symmetric CCN) with flip rate $\rho < \frac{1}{2}$,

$$\Pr(A = 1 \mid X = x) \;=\; (1 - 2\rho)\,\eta(x) + \rho, \tag{10}$$

a strictly increasing transform of $\eta(x)$. Minimizing a classification-calibrated surrogate on $A$ yields a scorer whose optimal threshold (shifted by $\rho$) implements the clean Bayes rule. In contrast, selecting RoIs directly via $A$ suffers from the conditional variance $\mathrm{Var}(A \mid X)$, leading to higher false positives/negatives, especially in low-margin regions.

**Implication for RoI prediction.** A RoI head trained to regress $A$ learns $\mathbb{E}[A \mid X]$, which (i) is order-preserving with $\eta(x)$ by the affine forms above, and (ii) enjoys reduced variance. Thresholding a learned predictor $\hat{h}(X)$ therefore yields more accurate and stable RoI proposals than thresholding raw attention $A$, matching our empirical gains. Our proposed *Label Assignment* strategy can be seen as a further refinement of this principle, where we structure the learning problem as a classification task on high-confidence tokens to make the training process even more robust to the noise in $A$.

## G  THE USE OF LARGE LANGUAGE MODELS

Throughout the preparation of this manuscript, we utilized a Large Language Model (LLM) as a general-purpose writing assistant to enhance the quality and clarity of our text. The LLM's role was strictly limited to that of a linguistic refinement tool. We used it to help polish our initial drafts by:

- Rephrasing sentences and paragraphs for improved readability, conciseness, and a more formal academic tone.
- Correcting grammatical errors, spelling, and punctuation.
- Improving the logical flow and transitions between sentences.
- Assisting with the generation of LaTeX code for tables based on our provided experimental data.

It is important to state that all core research ideas, the conceptualization of the proposed framework, the experimental design, the analysis of results, and the final scientific conclusions presented in this paper were conceived and articulated entirely by the human authors. The LLM did not contribute to the scientific ideation or the results of this work. The authors have reviewed, edited, and take full responsibility for all content in this manuscript.

