# OpenReview forum: "Catching the Details: Self-Distilled RoI Predictors for Fine-Grained MLLM Perception"
_ICLR.cc/2026/Conference — ICLR 2026 Poster_

### Official Review · Reviewer_jaiz · 2025-10-17

**Soundness:** 3
**Presentation:** 3
**Contribution:** 4
**Rating:** 8
**Confidence:** 4

**Summary:**

**Catching the Details: Self-Distilled ROI Predictors for Fine-Grained MLLM Perception** proposes a Self-Distilled ROI Predictor (SD-RPN) to address the high-res image perception problems. Unlike previous sft-based or training-free ROI-based methods, this paper pioneered using the filtered middle layer attention map as supervision signal, and trained several middle layers to serve as the SD-RPN. The overall idea is clear, elegant, and generalizable to other encoder+llm type of VLMs. The effectiveness of the proposed method is demonstrated via experiment, with solid and comprehensive growth across benchmarks.

**Strengths:**

1. The idea of Self-Distilled ROI Predictor is valuable, as it fully leverages the contextualization power of a VLM model, without relying on human-annotated data.
2. The training is not computation-efficient, and demonstrates consistent growth when data quantity is increased (Tab. 4b). This shows the potential of scaling up of this method.
3. The writing and illustration of the paper are excellent. The paper has expressive and clear figures, solid algorithms and detailed explanation for methods.

**Weaknesses:**

1. The experiments are limited to ROI methods and do not compare against other SoTA methods for high resolution. If authors could add some work such as Monkey, Token-Packer, Mini-Gemini, Honey-bee, LLaVA-NeXt, which have competitive numbers on the selected benchmarks such as DocVQA, the paper would be more solid.
2. The design of taking middle layers is not ablated. I have read Appendix A. While intuitively the middle layers should be a balance of visual details and global semantics, it should be proven, since using the middle layers can be much more expensive in computation compared with shallow layers.
3. (minor) The 1.6x slowdown does not support the claim of "light weight". Maybe try rephrasing the claim to be clearer.
4. (minor) The hyperparameters, like B and Sbg, Sfg thresholds, are not ablated.

References:
[1] Li, Z.; Yang, B.; Liu, Q.; Ma, Z.; Zhang, S.; Yang, J.; Sun, Y.; Liu, Y.; and Bai, X. 2023d. Monkey: Image resolution and text label are important things for large multi-modal models. arXiv preprint arXiv:2311.06607

[2] Li, W.; Yuan, Y.; Liu, J.; Tang, D.; Wang, S.; Zhu, J.; and Zhang, L. 2024a. Tokenpacker: Efficient visual projector for multimodal llm. arXiv preprint arXiv:2407.02392

[3] Li, Y.; Zhang, Y.; Wang, C.; Zhong, Z.; Chen, Y.; Chu, R.; Liu, S.; and Jia, J. 2024b. Mini-gemini: Mining the potential of multi-modality vision language models. arXiv preprint arXiv:2403.18814

[4] Cha, J.; Kang, W.; Mun, J.; and Roh, B. 2024. Honeybee: Locality-enhanced projector for multimodal llm. In IEEE CVPR, 13817–13827

[5] Liu, H.; Li, C.; Li, Y.; Li, B.; Zhang, Y.; Shen, S.; and Lee, Y. J. 2024a. LLaVA-NeXT: Improved reasoning, OCR, and world knowledge.

**Questions:**

1. Have authors considered ablating shallow layers? If not, is there a more convincing explanation?
2. Why are DocVQA and ChartQA significantly lower than the token compression methods?

I have a positive attitude towards this paper's contributions. I would maintain the high score if authors can address my concerns.

---

> ### Author Response · Authors · 2025-11-18
>
> ## **Q1. Comparison to SoTA High-Resolution Methods**
>
> We sincerely thank the reviewer for this insightful suggestion. We agree that a comparative analysis against other state-of-the-art (SoTA) high-resolution methods is crucial for positioning our work within the current landscape.
> In our revised manuscript, we have added a comprehensive comparison and discussion with the suggested SoTA methods in **Appendix C.4** section. Furthermore, we have **updated the Related Work (Section 2)** to properly cite these competitive approaches.
>
> To ensure a fair and transparent comparison, we present a detailed table below. This table highlights critical differences between methods—such as the base Large Language Model (LLM), the volume of training data, and the maximum input resolution—**all of which significantly influence performance**.
>
> | Method                  | LLM         | #Data  | Max Res.   | TextVQA val | OCRBench test | DocVQA test | ChartQA test |
> | :---------------------- | :---------- | :----- | :--------- | :---------- | :------------ | :---------- | :----------- |
> | LLaVA-1.5               | Vicuna-7B   | 1.2M   | 336x336    | 58.2#       | 31.4          | 22.2        | 18.1         |
> | LLaVA1.5+SD-RPN         | Vicuna-7B   | +152K  | 336x336    | 64.2#       | 37.0          | 35.1        | 20.1         |
> | Monkey                  | Qwen-7B     | 1.4M   | 896x896    | 67.6        | -             | 66.5        | 65.1         |
> | LLaVA-NeXT              | Vicuna-7B   | 1.3M   | 672x672    | 64.9*       | -             | -           | -            |
> | Mini-Gemini-HD          | Vicuna-7B   | 2.7M   | 536x1536   | 68.4*       | 456*          | 65.0*       | -            |
> | LLaVA-TokenPacker-HD    | Vicuna-7B   | 2.7M   | 1088x1088  | 68.0*       | 452*          | 60.2*       | -            |
> | DeepSeek-VL             | Vicuna-7B   | -      | 1024x1024  | 63.0        | 42.4          | 49.3        | 61.4         |
> | DeepSeek-VL+SD-RPN      | Vicuna-7B   | +152K  | 1024x1024  | 71.5        | 47.9          | 64.5        | 63.6         |
> | Qwen2.5-VL              | Qwen2.5-7B  | -      | 672x672    | 81.1        | 81.5          | 92.6        | 82.6         |
> | Qwen2.5-VL+SD-RPN       | Qwen2.5-7B  | +152K  | 672x672    | 83.5        | 82.9          | 94.4        | 85.5         |
>
> **Table Q1.** Comparison with SoTA High-Resolution Methods on Document and OCR Benchmarks. # denotes results obtained using OCR tokens, and * denotes results reported in the LLaVA-TokenPacker-HD paper. The "+152K" in the #Data column indicates the number of additional samples used for fine-tuning our SD-RPN on the corresponding baseline models.
>
> The results presented in Table Q1 reveal two key findings.
>
>  First, the overall performance of our method, like others, is strongly correlated with the capabilities of the underlying base MLLM. When applied to an earlier baseline such as LLaVA-1.5, our method's absolute scores on benchmarks like DocVQA are understandably lower than SoTA methods built on more advanced foundations, which often leverage richer training sets and higher input resolutions. This performance difference is primarily a reflection of the base model's capacity rather than a limitation of our enhancement module. We note that a direct comparison with Honey-bee was not accessible, as its performance on these specific benchmarks is not publicly reported.
>
> Second, our method's improvements are consistent across base models of varying strengths. When applied to DeepSeek-VL-7B, a more competitive baseline, our SD-RPN achieves performance that is on par with or slightly exceeds other leading methods. To further validate its effectiveness, we integrated SD-RPN with the powerful Qwen2.5-VL.
> Implementation details of this baseline have been updated in **Section 4** of our manuscript.
> Even on this state-of-the-art baseline, our method delivers consistent and significant gains across all benchmarks, using the identical, minimal fine-tuning recipe as with the other baselines. This highlights the primary advantage of our approach: SD-RPN offers a data-efficient pathway to enhance the fine-grained perception capabilities of MLLMs. Unlike methods that require extensive supervised fine-tuning on massive datasets, our self-distillation technique provides a scalable and resource-efficient solution for improving high-resolution understanding, which can readily adapt to the continuous advancements in base MLLMs.
>
> We thank the reviewer again for this constructive feedback, which has helped us significantly strengthen our paper.

---

> ### Author Response · Authors · 2025-11-18
>
> ## **Q2. RPN Layer ($B$) Ablation**
>
> We sincerely thank the reviewer for highlighting this critical point. Your insightful assessment that the middle layers should balance visual details and global semantics is completely correct. We apologize that this was not made sufficiently clear in our submission.
>
> The design of choosing the backbone layer depth, indicated as $B$ in the manuscript, directly addresses this question. This parameter was indeed ablated in **Table 4(a)** of the main paper and explained in the second paragraph of **Section 4.3**. This experiment, which varies $B$, directly investigates the impact of using shallower vs. deeper layers. We hope this clarification and the results below fully address this concern:
>
>
> | Setting | Throughput | OCRBench | TextVQA | POPE | V* | Ave. |
> | :--- | :--- | :--- | :--- | :--- | :--- | :--- |
> | Baseline | 1.0x | 31.4 | 46.1 | 85.9 | 50.3 | 53.4 |
> | B3R3 | **0.71x** | 33.9&uparrow;2.5 | 51.0&uparrow;4.9 | 87.0&uparrow;1.1 | 53.9&uparrow;3.6 | 56.5&uparrow;3.1 |
> | B6R3 | 0.63x | 34.4&uparrow;3.0 | 53.5&uparrow;7.4 | **87.7**&uparrow;1.8 | 57.1&uparrow;6.8 | 58.2&uparrow;4.8 |
> | B9R3 | 0.66x | 35.8&uparrow;4.4 | 56.3&uparrow;10.2 | 87.1&uparrow;1.2 | 59.7&uparrow;9.4 | 59.7&uparrow;6.3 |
> | B12R3 | 0.60x | 35.5&uparrow;4.1 | 57.8&uparrow;11.7 | 87.4&uparrow;1.5 | 67.0&uparrow;16.7 | 61.9&uparrow;8.5 |
> | **B15R3** | 0.62x | **37.0**&uparrow;5.6 | **58.7**&uparrow;12.6 | 87.1&uparrow;1.2 | **67.5**&uparrow;17.2 | **62.6**&uparrow;9.2 |
> | B18R3 | 0.52x | 35.5&uparrow;4.1 | 57.8&uparrow;11.7 | 87.3&uparrow;1.4 | 66.0&uparrow;15.7 | 61.7&uparrow;8.3 |
>
> We conduct an ablation study on the number of frozen backbone layers (B) that the RPN is built upon, keeping the number of trainable RPN layers fixed. We observe a clear trend in performance: as the backbone depth increases, the average performance steadily improves, reaching its peak with the B15R3 configuration, which achieves a 9.2\% gain over the baseline. Beyond this point, performance begins to decline.
> The relationship between backbone depth and efficiency is more nuanced. While the inference cost of the RoI prediction stage is positively correlated with the number of backbone layers, the overall throughput is not monotonic. This is because a more precise RoI prediction, often produced by a deeper backbone, can reduce the number of irrelevant or noisy visual tokens that are passed to the second, more costly generation stage. This interplay between the increasing cost of the first stage and the potentially decreasing cost of the second stage explains the fluctuating throughput values.
>
> ## **Q3. Incorrect "Lightweight" Claim**
> We sincerely thank the reviewer for this valuable feedback. To avoid any misleading expressions, we have revised the manuscript to remove the "lightweight" claim. We now more precisely describe our method's efficiency in terms of its better performance-to-throughput trade-off and its more efficient RoI prediction stage compared to competing two-stage approaches. Thank you once again for helping us make this manuscript clearer and more precise.
>
> ## **Q4: Ablation Study on Pseudo-Label Thresholds**
>
> We deeply appreciate the reviewer's insightful comment on the sensitivity of our method to the $\tau_{fg}$ and $\tau_{bg}$ hyperparameters. We agree that this is a critical validation, and we acknowledge this was an oversight in our initial submission. To address this, we have now conducted a detailed ablation study to identify the optimal threshold settings. The results are as follows:
>
> | $\tau_{fg}, \tau_{bg}$ | OCRBench | TextVQA | POPE | V\* | Ave. |
> | :--- | :---: | :---: | :---: | :---: | :---: |
> | Baseline | 31.4 | 46.1 | 85.9 | 50.3 | 53.4 |
> | 0.10, 0.10 | 35.1 | 57.4 | 87.4 | 65.4 | 61.3 |
> | 0.15, 0.10 | 36.3 | 57.9 | 87.2 | 67.5 | 62.2 |
> | 0.20, 0.10 | 37.0 | 58.7 | 87.1 | 67.5 | 62.6 |
> | 0.25, 0.10 | 37.0 | 58.6 | 86.7 | 68.1 | 62.6 |
> | 0.30, 0.10 | 36.6 | 58.1 | 86.9 | 64.9 | 61.6 |
> | 0.20, 0.075 | 36.7 | 58.4 | 86.9 | **70.7** | 63.2 |
> | **0.20, 0.05** | **37.3** | **58.8** | 87.2 | **70.7** | **63.5** |
> | 0.20, 0.03 | 37.8 | 58.7 | 87.6 | 68.6 | 63.2 |
>
> Surprisingly, the new optimal hyper-parameter, $\tau_{fg}=0.20, \tau_{bg}=0.05$, outperforms our previous default setting ($\tau_{fg}=0.20, \tau_{bg}=0.10$) by nearly 1% on average over these four diverse benchmarks!
>
> We wish to express our sincere gratitude. It is your suggestion that pushed the performance boundary of our work. The corresponding results have been updated in **Table 5** of the manuscript, and we have also updated the main results in **Table 1** and **Table 2** to reflect this stronger default.

---

> ### Author Response · Authors · 2025-11-18
>
> ## **Q5: Performance Discrepancy**
> We sincerely thank the reviewer for this crucial question, which prompted us to conduct a deeper analysis against SOTA high-resolution methods. We respectfully clarify the performance discrepancy as follows.
>
> The performance gap seen on DocVQA and ChartQA when using the LLaVA-1.5 baseline is not primarily a comparison of RoI-selection (our method) vs. token-compression. Rather, it is a reflection of **fundamental differences in the training data used by the base models**. The SOTA methods cited (e.g., Monkey, Mini-Gemini-HD, LLaVA-TokenPacker-HD) explicitly incorporate the training splits of these text-rich benchmarks (e.g. DocVQA, ChartQA) into their supervised fine-tuning stage, as detailed in their respective papers (Monkey: Tbl. 1, Mini-Gemini: Sec. 3.3, TokenPacker: Sec. 4.2). In contrast, the LLaVA-1.5 baseline we used was not fine-tuned on this data, resulting in its significantly lower baseline performance.
>
> Our method, SD-RPN, is designed as an efficient perception module that does not fine-tune the parameters of the base MLLM. Its effectiveness is therefore heavily reliant on the foundational capabilities of the model it is built upon. While SD-RPN substantially improves the LLaVA-1.5 baseline (e.g., +12.4 on DocVQA and +2.0 on ChartQA), it cannot by itself bridge the massive performance gap created by this data disparity. **Crucially, this is validated by our results on stronger baselines.** When SD-RPN is applied to **DeepSeek-VL-7B** and **Qwen2.5-VL-7B** models that are competitive on these benchmarks—our method's performance scales accordingly (as shown in Table.Q1). On DeepSeek-VL, our method (+SD-RPN) closes the gap, achieving competitive or better performance. On Qwen2.5-VL, which is already a SOTA baseline, our method further improves its performance, achieving 94.4 on DocVQA test split and 85.5 on ChartQA test split. These results demonstrate that the performance discrepancy is a factor of the baseline model's training data, not an inherent limitation of our RoI method.
>
>
> We hope this response adequately addresses the reviewer's insightful concern.

---

> > ### Comment · Reviewer_jaiz · 2025-11-19
> > **Authors clearly addressed my concerns**
> >
> > Thanks for the clarification and experiments authors have done. My questions have been clearly addressed. As a researcher on VLM architecture, I resonate with authors that performance, especially OCR tasks, relies heavily on the backbone and training data.
> >
> > With the added experiments, revised phrasing and detailed explanation, I believe the updated paper can contribute meaningfully to the research community in VLM. Therefore, I keep the score of 8, and support the acceptance of this paper.

---

> > > ### Author Response · Authors · 2025-11-25
> > >
> > > Dear Reviewer jaiz,
> > >
> > > As the discussion phase concludes soon, we would like to sincerely thank you for your time and effort in reviewing our paper.
> > >
> > > We truly appreciate your positive assessment and recognition of our work. Your insightful comments have been very valuable to us, and we are grateful for your support.
> > >
> > > Best regards,
> > > The Authors

---

### Official Review · Reviewer_KUum · 2025-10-25

**Soundness:** 3
**Presentation:** 3
**Contribution:** 3
**Rating:** 6
**Confidence:** 3

**Summary:**

This paper introduces a novel annotation-free framework termed the Self-Distilled Region Proposal Network (SD-RPN), designed to address the challenge that MLLMs face in comprehending fine-grained text or objects within high-resolution images. It circumvents the limitations of existing approaches: training-based methods that rely on large-scale annotated datasets, and training-free techniques that suffer from computational inefficiency and lower accuracy. The proposed approach transforms the inherently noisy attention maps within MLLMs into high-quality pseudo-RoI labels, which are subsequently used to train a lightweight RPN. Furthermore, the weights from middle layers of the MLLM are used to initialize the RPN, enhancing its performance. Empirical results demonstrate the high effectiveness of the method, achieving substantial absolute accuracy gains (over 10%) on benchmarks such as TextVQA and DocVQA when integrated into existing MLLMs.

**Strengths:**

1. This paper introduces a self-distillation framework for training a lightweight RPN, resolving the difficult trade-off between training-based methods, which require costly human annotations, and training-free methods, which are often inefficient and inaccurate.
2. The pseudo-label generation pipeline proposed is clear and feasible, as it involves (i) identifying and removing "sink tokens" and (ii) adopting a selective classification strategy that labels only high-confidence foreground/background regions.
3. The paper validates the effectiveness of the proposed approach through a wide range of benchmarks.

**Weaknesses:**

1. This paper introduces thresholds in stages such as pseudo-label generation and RoI prediction. However, the authors do not justify the rationale behind these specific threshold values, nor do they explore how variations in these thresholds might affect model performance.
2. The paper seems to lack an in-depth analysis regarding the sources of "noise" during pseudo-label generation. For instance, it does not adequately address the types of noise involved, how such noise is generated, or what kinds of images or tasks are prone to producing noisy labels.
3. The qualitative examples presented in Figure 5 effectively showcase the successes of SD-RPN. However, a comprehensive evaluation should also include an analysis of its failure modes. The paper would be strengthened by showing examples where SD-RPN fails and diagnosing the cause.

**Questions:**

1. How were the specific threshold values (e.g., 0.2 and 0.1 for foreground/background definition) determined?
2. Which types of images or questions are more prone to introducing noise in the attention maps?

---

> ### Author Response · Authors · 2025-11-18
>
> ## **Q1. Threshold Rationale**
>
> We deeply appreciate the reviewer's insightful comment on the sensitivity of our method to the $\tau_{fg}$ (foreground) and $\tau_{bg}$ (background) hyperparameters. We respectfully offer the following two-part explanation for these values
>
> **1. Initial Rationale (from Manuscript Figure 3):** Our initial threshold values were not arbitrary but were guided by an empirical analysis of the attention maps, which is detailed in **Figure 3** of the manuscript. As shown in that figure, we analyzed the correlation between attention scores and ground-truth bounding boxes. We found that: High-attention scores (e.g., > 0.2) had a ~40% probability of being inside the GT box; Low-attention scores (e.g., < 0.1) had only a ~10% probability. Based on this clear statistical pattern, we set our initial default thresholds to $\tau_{fg}=0.20$ and $\tau_{bg}=0.10$.
>
> **2. Full Ablation Study (New Results):**
> We completely agree with the reviewer that a detailed ablation study is critical for validation, and we thank you for motivating this deeper analysis. We have conducted a full study on these thresholds, with the results now presented in **Table 5**:
>
> | $\tau_{fg}, \tau_{bg}$ | OCRBench | TextVQA | POPE | V\* | Ave. |
> | :--- | :---: | :---: | :---: | :---: | :---: |
> | Baseline | 31.4 | 46.1 | 85.9 | 50.3 | 53.4 |
> | 0.10, 0.10 | 35.1 | 57.4 | 87.4 | 65.4 | 61.3 |
> | 0.15, 0.10 | 36.3 | 57.9 | 87.2 | 67.5 | 62.2 |
> | 0.20, 0.10 | 37.0 | 58.7 | 87.1 | 67.5 | 62.6 |
> | 0.25, 0.10 | 37.0 | 58.6 | 86.7 | 68.1 | 62.6 |
> | 0.30, 0.10 | 36.6 | 58.1 | 86.9 | 64.9 | 61.6 |
> | 0.20, 0.075 | 36.7 | 58.4 | 86.9 | **70.7** | 63.2 |
> | **0.20, 0.05** | 37.3 | **58.8** | 87.2 | **70.7** | **63.5** |
> | 0.20, 0.03 | **37.8** | 58.7 | **87.6** | 68.6 | 63.2 |
>
>
> This study confirms that our initial default (`0.20, 0.10`) was a strong choice (62.6% average). However, it also revealed a new, **optimal configuration of $\tau_{fg}=0.20, \tau_{bg}=0.05$**. This new setting achieves a **63.5%** average performance, surpassing our original default by nearly 1%.
>
> We wish to express our sincere gratitude. Your suggestion directly pushed us to find this improved configuration, which has measurably enhanced our method's performance. We have updated **Table 5** with this full study and have also updated the main results in **Table 1** and **Table 2** to reflect this stronger default.

---

> ### Author Response · Authors · 2025-11-18
>
> ## **Q2. In-Depth Analysis on the Sources of Noise**
>
> We thank the reviewer for this insightful question and we respectfully offer the following explanation. A full, novel investigation into the fundamental *generation* of attention noise is a complex research area in itself. Our work, in contrast, builds upon established findings in the literature to **characterize and actively mitigate** this noise.
> Our paper identifies and addresses two primary types of noise in the attention maps, which we use to generate our pseudo-labels:
>
> 1.  **Type 1: Sink Tokens.** As the reviewer notes, we build on recent studies that identify "sink tokens". These are visual tokens that attract high attention despite being semantically irrelevant to the grounded object. This appears to be a *structural artifact* of the Transformer architecture, and these tokens can be reliably identified by the high L2-norm of their feature vectors. Our "Remove Sink Tokens" step in the pipeline directly targets this known, well-defined source of noise.
>
> 2.  **Type 2: Ambiguous Foreground/Background Signals.** This is the more common "noise" our paper focuses on. As illustrated in Figure 2 and described in Section 3.2, these are "noisy maps" that exhibit "erroneously high attention in background areas and incomplete activation across the true foreground object". This type of noise is *semantic* and *task-dependent*.
>
> To directly address the reviewer's question about which images or tasks are prone to this noise, we performed an additional empirical study. We have now complemented the manuscript with a new discussion and visualizations in **Appendix C.3** and **Figure.7**. We summarize these key findings as follows:
>
> * **Cluttered Scenes & Dense Text:** Images with many objects or dense text (e.g., the first sample in Figure.7) are highly prone to this. The model's attention can be "fraught with noise"  as it diffuses across multiple irrelevant-but-plausible regions.
> * **Abstract or Reasoning-Heavy Queries:**  In practice, we find that a simple query (e.g., "find the cat") tends to produce clean attention. However, many VQA queries are abstract, such as the second sample in Figure.7. For such queries, the attention map reflects the model's complex *reasoning process*, which can appear noisy as it attends to multiple pieces of related (but not all relevant) information before settling on an answer.
> * **Fine-Grained Details:**  The last type of noise is related to *incomplete activation*, where the attention map only weakly or partially covers the true region of interest, as shown in the last sample of Figure.7.
>
> We thank the reviewer again for this valuable question. It motivated us to conduct a deeper analysis, and we hope our response above fully addresses this concern.
>
> ## **Q3. Failure Cases**
>
> We thank the reviewer for this highly constructive comment. Analyzing failure cases where the baseline succeeds is indeed essential for understanding the limitations of our approach and guiding future improvements.
> We have prepared a detailed failure case analysis, as summarized in the **Figure.9** of appendix and text below. These examples show that our method's failures can be broadly attributed to three main reasons:
>
> 1.  **Incomplete Activation:** The top-left "cup" case is a representative example. Our RPN correctly identifies the measuring cup but only activates on a small portion of it. This partial RoI provides incomplete visual information to the MLLM, leading to an erroneous prediction ("1") where the baseline, seeing the global context, correctly identifies "5".
>
> 2.  **Localization Error:** The top-right "book" example illustrates this. The RPN is supposed to highlight the book's cover to read the name ("Steve") but erroneously highlights an unrelated region. Given this misleading visual cue, the MLLM fails and guesses "Doug".
>
> 3.  **Spatial Relationship Error:** The bottom row shows a more subtle but critical limitation. In these cases, the RPN highlights *all relevant objects* correctly (e.g., the "bowl" and "faucet" or the "seesaw" and "vehicle"). However, the MLLM still produces the wrong spatial answer. This suggests that even with the correct objects localized, our current two-stage approach struggles to preserve and reason about the *precise spatial relationships* (e.g., "left of") between objects once they are processed as fine-grained crops. The baseline, operating on a single global image, retains this spatial context more effectively.
>
> This analysis is invaluable, and we have added this figure and discussion to the paper. It highlights a clear direction for future work: developing methods to better integrate fine-grained local details without losing essential global spatial context. Thank reviewer again for this constructive comment!

---

> > ### Comment · Reviewer_KUum · 2025-11-24
> >
> > Thank you for the detailed response. The rebuttal has successfully clarified the points I raised. I will maintain my positive score.

---

> > > ### Author Response · Authors · 2025-11-25
> > >
> > > Dear Reviewer KUum,
> > >
> > > As the discussion phase concludes soon, we would like to sincerely thank you for your time and effort in reviewing our paper.
> > >
> > > We truly appreciate your positive assessment and recognition of our work. Your insightful comments have been very valuable to us, and we are grateful for your support.
> > >
> > > Best regards,
> > > The Authors

---

### Official Review · Reviewer_ScXn · 2025-10-29

**Soundness:** 3
**Presentation:** 4
**Contribution:** 3
**Rating:** 4
**Confidence:** 3

**Summary:**

This paper proposes a method to enhance the visual comprehension capability of a multimodal large language model (MLLM) by training a region-proposal network, which is initialized from a subset of the MLLM's parameters. The training data for the region proposals are derived from the attention maps of intermediate layers in the MLLM. To obtain more precise and less noisy region-of-interest (ROI) targets, the authors introduce several techniques, such as removing sink tokens and dynamically assigning labels to instances.

With the trained region-proposal network, the MLLM can perform two-stage inference: in the first stage, the model predicts ROIs using the proposal network; in the second stage, it processes a high-resolution version of the selected ROI to enable more detailed visual information extraction. The authors conduct extensive experiments to validate the effectiveness of the proposed method, and provide ablation studies to assess the contribution of each design component.

**Strengths:**

* The overall presentation of the paper is strong. The manuscript is well-written, logically structured, and easy to follow.

* Extensive experiments are conducted to demonstrate the superiority of the proposed method across multiple MLLM benchmarks.

* The method achieves notable performance improvements, particularly on OCR-related tasks, highlighting its strengths in fine-grained visual understanding.

* The ablation studies and visualizations provide clear evidence of the effectiveness of the key components in the proposed approach.

**Weaknesses:**

* I question the necessity of extracting ROIs from attention maps, as they can be noisy and spatially imprecise. Given the availability of large-scale object detection datasets (e.g., OpenImages) and powerful MLLMs capable of generating captions, it may be feasible to construct a dataset with high-quality region-text annotations. It would be helpful for the authors to discuss whether their approach offers any advantages—such as cost, scalability, or task-specific alignment—over using such manually or model-annotated datasets.

* The authors propose several strategies for utilizing the predicted bounding boxes. However, a concern arises: if only the cropped ROI is used for answering the current question, could this lead to a loss of global visual context? While the global scene may appear irrelevant to the immediate query, it might provide valuable information for subsequent questions in a multi-turn dialogue. In such cases, does the proposed two-stage inference risk discarding useful contextual cues, potentially harming performance over extended conversations?

* The RPN is designed to predict ROIs based on the provided textual context. However, using a portion of an MLLM with billions of parameters for this purpose seems computationally heavy. Since the goal resembles that of open-set grounding models, it would be insightful to compare against lightweight alternatives such as fine-tuned Grounding DINO or YOLO-World. Could such models achieve comparable ROI localization performance with significantly lower computational overhead?

* The standalone performance of the proposed RPN remains unclear. There are no dedicated benchmarks or quantitative metrics reported to evaluate its accuracy in ROI prediction. Could the authors clarify how they validated the quality of the generated proposals during development? For instance, were human evaluations or proxy metrics used?

* I noticed that some visualization results are provided in the supplementary material. Including a few representative examples in the main paper—such as attention maps, predicted ROIs, and corresponding model outputs—would greatly enhance the reader’s understanding of the method’s working mechanism and effectiveness.

* I am also interested in potential failure cases. Are there instances where the baseline MLLM answers correctly but the proposed method fails? Analyzing such cases could shed light on the limitations of the current approach and guide future improvements.

**Questions:**

Please see weaknesses.

---

> ### Author Response · Authors · 2025-11-18
>
> ## **Q1. Justification for Self-Distillation.**
>
> We deeply thank the reviewer for this detailed and insightful question. We value this opportunity for further clarification. To support our response, we have trained SD-RPN on the **Qwen2.5-VL-7B** baseline using the same recipe as for LLaVA and DeepSeek-VL, which provides new experimental data to directly address this concern.
> Implementation details have been updated in **Section 4** of our manuscript.
>
> Our self-distillation approach offers distinct advantages over using external annotated datasets in terms of **task-specific alignment**, **data efficiency**, and **preservation of model generality**.
> Manually annotated datasets, while accurate, are often limited to predefined categories, which can constrain generalizability. For example, training an RPN on a general-purpose "object" dataset would teach it to find *all* objects (the fridge, a sign, a handle, etc.), not the *specific, task-relevant* one required by the prompt.
> There is indeed recent work, such as **VILA-HD[1]**, that **utilizes SAM and a powerful MLLM (Qwen2-VL) to create large-scale region-text pairs for training models to identify RoIs.** We adopt this work as an example in this type for comparison.
> However, this type of method suffers from several critical issues:
>
> **1. Data Quality and Efficiency.**
> Region-text pairs sourced from a "teacher" MLLM are limited by that teacher's capabilities. For example, using a synthetic dataset from Qwen2-VL to train the newer Qwen2.5-VL model would not unleash the full power of the Qwen2.5-VL model. Furthermore, this synthetic dataset may be *out-of-distribution* compared to the model's original pre-training data, requiring far more samples to achieve a similar fine-grained perception capability. This is evidenced by VILA-HD, which utilizes over **282M** region-text pairs for training while its overall performance still lags behind the Qwen2-VL model used to create its dataset (as shown below).
> In contrast, our method distills knowledge from the MLLM's *own* attention mechanism. The "teacher" (MLLM attention) and "student" (RPN) are **inherently aligned**, sharing the same internal "language." Therefore, it takes far fewer samples to converge. The following table, with new results for Qwen2.5-VL, demonstrates this. VILA-HD's performance (sourced from its original paper) lags behind its teacher, while our method (using only **152K** QA pairs) significantly outperforms its stronger Qwen2.5-VL baseline.
>
> **Benchmark comparison of SD-RPN (Ours) applied to Qwen2.5-VL-7B against its baseline and the VILA-HD method.**
>
> | Methods | Res. | ChartQA | DocVQA | InfoVQA | OCRBench | V*Bench | TextVQA |
> | :--- | :---: | :---: | :---: | :---: | :---: | :---: | :---: |
> | Qwen2-VL-7B | 3584 | 83.0 | 94.5 | 76.5 | 86.6 | 71.0 | 84.3 |
> | VILA-HD-4K-8B | 3780 | 86.1 | 91.7 | 65.3 | 77.4 | 71.2 | 77.9 |
> | Qwen2.5-VL-7B | **672** | 82.6 | 92.0 | 70.2 | 81.5 | 64.4 | 81.1 |
> | Qwen2.5-VL-7B+Ours | **672** | 85.5 | 93.6 | 76.9 | 82.9 | 77.5 | 84.5 |
>
> **2. Domain Constraint and Generality.**
> Powerful MLLMs like Qwen2.5-VL are pre-trained on trillions of tokens. Creating a synthetic dataset, even with millions of samples, introduces a significant domain gap. Fine-tuning the entire model on this synthetic data will **inevitably harm the generality** of the original pre-trained model.
> In contrast, our SD-RPN is built upon the original model, **freezing the pre-trained parameters** and training only the RPN. This approach **fully preserves the original generality** of the pre-trained model. The self-distillation simply makes the model's inherent fine-grained ability *explicit*. This claim is well-supported by the V-Star benchmark results: VILA-HD achieves performance similar to its teacher, whereas our method achieves a much larger gain over its (stronger) baseline, demonstrating superior generalization.
>
> **3. Training Cost.**
> As a result of this data efficiency and model design, training SD-RPN on Qwen2.5-VL-7B cost only **12 GPU hours** on NVIDIA A6000 GPUs, making it a highly practical solution for scenarios with limited computational budgets.

---

> ### Author Response · Authors · 2025-11-18
>
> ## **Q2. Loss of Global Context**
>
> We appreciate the reviewer raising this important concern. This is a critical point, and we respectfully clarify that our method **does not lose or discard the global visual context**. The reviewer's concern may stem from a misunderstanding of our two-stage inference process, and we apologize for any misleading expressions in the paper.
> Our method is designed to be *augmentative*, not *replacement*. The high-resolution RoI tokens are **inserted into the token sequence alongside the original full-image tokens**.
> The simplified inference process for a single QA-pair is:
>
> 1.  **Stage 1 (RoI Prediction):** The system prompt, the *entire* low-resolution image, and the user query are passed through the RPN (the first `B+R` layers of the MLLM) to predict the RoI.
> 2.  **Stage 2 (Final Generation):** The model processes an *augmented* sequence:
>     `[Sys Prompt] + [Original Image Tokens] + [High-Res RoI Tokens] + [Question Tokens]`
>
> Critically, the `[Original Image Tokens]` are **never removed**. The model *always* has access to the full global scene. Furthermore, thanks to the KV-Cache, we **only need to re-forward the *new* RoI and question tokens** through the first `B` frozen layers, making this augmentation highly efficient.
>
> Therefore, the model has access to *both* the complete global context (from the original tokens) and the fine-grained details (from the new RoI tokens) for all subsequent layers, fully addressing the concern for multi-turn dialogue.
>
>
> ## **Q3. Computational Cost of the RPN**
>
> We thank the reviewer for raising this practical concern.
> It may seem counter-intuitive, but our SD-RPN is, in fact, significantly **more computationally efficient** than using an external grounding model like Grounding-DINO or YOLO-World. The reason is that our RPN **re-uses computations** that the MLLM must perform anyway. Below is a brief comparison of these two types of method:
>
> 1.  **Redundant Computation (External Models):** An external model (e.g., Grounding-DINO) would require a *separate, full* image encoding pass, which is computationally expensive.
> 2.  **Our Approach (Feature Re-use):** Our RPN attaches to the MLLM's middle layers. It operates on features that are *already computed* during the MLLM's initial forward pass (as described in Q2). The *only* extra cost is passing these features through the 3 RPN layers and forwarding the question token twice, which is a minimal addition.
>
> To provide a concrete analysis, we benchmarked our method (on LLaVA-1.5-7B) against a Grounding-DINO-Base model on a NVIDIA A6000 GPU. The "Grounding-DINO-B (1Box)" method crops one sub-image using the single most confident bounding box, while "(2Box)" crops two sub-images using the top 2 boxes.
>
> | Method | RPN Time (ms) | V* Eval Time (s) | OCRBench | TextVQA | POPE | V* | Ave. |
> | :--- | :---: | :---: | :---: | :---: | :---: | :---: | :---: |
> | **Ours (SD-RPN)** | **10.3** | **73** | **37.0** | **58.7** | **87.1** | **67.5** | **62.6** |
> | Grounding-DINO-B (2Box) | 231.2 | 248 | 32.5 | 54.5 | 82.4 | 62.3 | 57.6 |
> | Grounding-DINO-B (1Box) | 231.2 | 113 | 32.5 | 52.6 | 86.9 | 56.5 | 57.1 |
>
> As shown in the table, our RPN is **over 20x faster** (10.3ms vs 231.2ms) because it avoids a redundant, full-image encoding. Besides, our method is also **significantly more accurate** (62.6% vs 57.6% avg.). This is because our RPN is part of the MLLM and has access to its *contextual understanding* of the query. External models are excellent at finding *literal objects* (e.g., "the bottom man") but fail at abstract, reasoning-based queries that don't name a specific object, such as the example in last row of Fig. 7 ("What is the Budget Estimate for...?"). Our RPN, trained on the MLLM's internal attention, excels at this. We have supplemented the manuscript with this discussion and these results in **Appendix C.2**, and we sincerely hope this response addresses your concern.
>
> Finally, we wish to express our deepest gratitude to the reviewer for this valuable comment. It gave us the opportunity to rethink the inference stage of our work, leading us to find optimization techniques—such as reusing the KV cache for system and image tokens—that we had overlooked in our original manuscript.

---

> ### Author Response · Authors · 2025-11-18
>
> ## **Q4. Standalone Performance and Validation of the RPN**
>
> We thank the reviewer for this insightful question regarding the RPN's standalone validation. The reviewer is correct that we do not report IoU or other traditional detection metrics, and this is a deliberate choice based on the nature of our task.
>
> **The "Ground Truth" Problem.**
> For the VQA datasets we use, there are **no ground-truth (GT) bounding boxes** for the RoIs. Unlike object detection, our queries are often abstract and highly contextual (e.g., "What is the Budget Estimate for..." in Fig.7 , "What city is written above the tire?" in Fig.5). A GT bounding box for "Budget Estimate" simply does not exist.
>
> **End-to-End Performance as the Proxy Metric.**
> Given the lack of GT labels, the **end-to-end performance on the downstream VQA task serves as the primary and most meaningful quantitative metric** for the RPN's quality. If the RPN provides high-quality, relevant RoIs, the MLLM's final accuracy will improve. If it provides noisy or incorrect RoIs, the performance will drop. This evaluation setting is also **adopted from prior work**, such as ViCrop[2], which inspired our approach.
>
> Our **Ablation Study in Table 3** was designed to be this *exact* quantitative validation.
>
> * **Baseline**: The baseline LLaVA-1.5-7B model (row 0) achieves a **53.4%** average score.
> * **Direct Use of Noisy Attention**: If we simply *use* the raw, noisy attention map (our "teacher" signal) to select the RoI (row 1), the score only improves to **56.9%**. This confirms the reviewer's suspicion that raw attention is noisy.
> * **Our Full Method**: When our **SD-RPN is trained on the *denoised* pseudo-labels** (row 5), the performance jumps to **62.6%**.
>
> This **+9.2% absolute improvement** over the baseline (and +5.7% over using the noisy signal directly) is the quantitative proof that our RPN is *not* just copying the noisy attention. It is successfully learning a precise, high-quality localization function from the *denoised* signal we generate.
>
> **Qualitative Validation (Human Evaluation Proxy).**
> As a proxy for human evaluation, we provided **extensive qualitative visualizations in Figure 5 and Figure 7**. These examples show *how* the RPN is working. In these figures, the RPN successfully localizes abstract or fine-grained information——which directly leads to the correct final answer where the baseline fails. This visual evidence serves to validate the *quality* of the RoIs our RPN generates.
>
> In short, while a standalone IoU-style benchmark isn't feasible, we validated the RPN's quality quantitatively through the *significant* downstream gains in our ablation study (Table 3) and qualitatively through visualizations (Fig 5, 7).
>
>
> ## **Q5. Visualization in the Main Paper**
>
> We thank the reviewer for this excellent suggestion. We completely agree that including representative visualizations in the main paper is crucial for demonstrating the method's working mechanism and effectiveness.
> We have included such a figure in the main manuscript. **Figure 5** in page 9 provides exactly this: it presents challenging qualitative examples, showing the **RoI Heatmap** (the predicted attention map) and the **Mixed Image** (which highlights the final high-confidence region used for fine-grained perception), alongside the baseline and our model's final outputs. We hope this figure directly addresses your suggestion and clearly illustrates how our SD-RPN operates to achieve the correct answer.

---

> ### Author Response · Authors · 2025-11-18
>
> ## **Q6. Analysis of Failure Cases**
>
> We thank the reviewer for this highly constructive comment. Analyzing failure cases where the baseline succeeds is indeed essential for understanding the limitations of our approach and guiding future improvements.
> We have prepared a detailed failure case analysis, as summarized in the **Figure.9** of appendix and text below. These examples show that our method's failures can be broadly attributed to three main reasons:
>
> 1.  **Incomplete Activation:** The top-left "cup" case is a representative example. Our RPN correctly identifies the measuring cup but only activates on a small portion of it. This partial RoI provides incomplete visual information to the MLLM, leading to an erroneous prediction ("1") where the baseline, seeing the global context, correctly identifies "5".
>
> 2.  **Localization Error:** The top-right "book" example illustrates this. The RPN is supposed to highlight the book's cover to read the name ("Steve") but erroneously highlights an unrelated region. Given this misleading visual cue, the MLLM fails and guesses "Doug".
>
> 3.  **Spatial Relationship Error:** The bottom row shows a more subtle but critical limitation. In these cases, the RPN highlights *all relevant objects* correctly (e.g., the "bowl" and "faucet" or the "seesaw" and "vehicle"). However, the MLLM still produces the wrong spatial answer. This suggests that even with the correct objects localized, our current two-stage approach struggles to preserve and reason about the *precise spatial relationships* (e.g., "left of") between objects once they are processed as fine-grained crops. The baseline, operating on a single global image, retains this spatial context more effectively.
>
> This analysis is invaluable, and we have added this figure and discussion to the paper. It highlights a clear direction for future work: developing methods to better integrate fine-grained local details without losing essential global spatial context. Thank reviewer again for this constructive comment!
>
>
> [1]. Shi, Baifeng, et al. "Scaling Vision Pre-Training to 4K Resolution." CVPR 2025.
>
> [2]. Jiarui Zhang, et al. Mllms know where to look: Training-free perception of small visual details with multimodal llms. ICLR 2025.

---

> ### Author Response · Authors · 2025-11-25
>
> Dear Reviewer ScXn,
>
> We are writing to respectfully invite you to review our responses and the new experimental results we have provided.
>
> As the author-reviewer discussion phase is concluding soon, we are keen to ensure that our revisions have adequately addressed your concerns. We would greatly appreciate any further feedback or confirmation you might have before the deadline.
>
> Thank you again for your time and valuable comments.
>
> Best regards,
> The Authors

---

### Official Review · Reviewer_2wEb · 2025-11-01

**Soundness:** 2
**Presentation:** 3
**Contribution:** 3
**Rating:** 4
**Confidence:** 4

**Summary:**

This paper proposes a Self-Distilled Region Proposal Network (SD-RPN). The pipeline first converts the response-to-image attention from the MLLM into high-quality pseudo RoI labels through two key steps: removing sink tokens and assigning labels. Then, the RPN is initialized using the corresponding layers of the MLLM. It collects hidden-state sequences from the second-to-last layer and uses the corresponding token sequences to predict an RoI map. Finally, the RPN is trained with self-distillation, constrained by these pseudo RoI labels. Trained on only a few question-answer pairs, the method achieves great accuracy and efficiency improvement on unseen benchmarks.

**Strengths:**

1. Annotation-free RoI distillation mechanism that elegantly leverages internal attention for supervision.
2. Improvement on accuracy and efficiency with comprehensive experimental validation across diverse benchmarks.
3. Clear motivation and insight, especially the analysis of noisy attention and the role of sink tokens.

**Weaknesses:**

1. The pseudo-label generation method relies on fixed thresholds (e.g., $τ_{fg}$=0.2) to distinguish foreground, background, and ignored regions. These thresholds directly affect the quality of the pseudo labels, but the paper lacks experimental validation with different parameter settings across multiple datasets.
2. Lack of experiments on very recent MLLMs (e.g., Qwen2.5-VL).
3. Efficiency improvement can be further quantified in wall-clock latency and GPU cost.

**Questions:**

1. SD-RPN was tested on LLaVA and DeepSeek-VL. How does it transfer to other architectures such as Qwen2.5-VL?
2. What is the performance of SD-RPN if pseudo-labels were replaced by ground-truth bounding boxes? Is it the upper bound of SD-RPN?
3. What is the performance of SD-RPN compared to thinking with image methods?
4. Why was BCE chosen over other robust alternatives (e.g., focal loss or IoU-based loss) given the heavy class imbalance between foreground and background tokens?

---

> ### Author Response · Authors · 2025-11-18
>
> ## **Q1: Ablation Study on Pseudo-Label Thresholds**
>
> We deeply appreciate the reviewer's insightful comment on the sensitivity of our method to the $\tau_{fg}$ and $\tau_{bg}$ hyperparameters. We agree that this is a critical validation, and we acknowledge this was an oversight in our initial submission. To address this, we have now conducted a detailed ablation study to identify the optimal threshold settings. The results are as follows:
>
> | $\tau_{fg}, \tau_{bg}$ | OCRBench | TextVQA | POPE | V\* | Ave. |
> | :--- | :---: | :---: | :---: | :---: | :---: |
> | Baseline | 31.4 | 46.1 | 85.9 | 50.3 | 53.4 |
> | 0.10, 0.10 | 35.1 | 57.4 | 87.4 | 65.4 | 61.3 |
> | 0.15, 0.10 | 36.3 | 57.9 | 87.2 | 67.5 | 62.2 |
> | 0.20, 0.10 | 37.0 | 58.7 | 87.1 | 67.5 | 62.6 |
> | 0.25, 0.10 | 37.0 | 58.6 | 86.7 | 68.1 | 62.6 |
> | 0.30, 0.10 | 36.6 | 58.1 | 86.9 | 64.9 | 61.6 |
> | 0.20, 0.075 | 36.7 | 58.4 | 86.9 | **70.7** | 63.2 |
> | **0.20, 0.05** | **37.3** | **58.8** | 87.2 | **70.7** | **63.5** |
> | 0.20, 0.03 | 37.8 | 58.7 | 87.6 | 68.6 | 63.2 |
>
> Surprisingly, the new optimal hyper-parameter, $\tau_{fg}=0.20, \tau_{bg}=0.05$, outperforms our previous default setting ($\tau_{fg}=0.20, \tau_{bg}=0.10$) by nearly 1% on average over these four diverse benchmarks! We wish to express our sincere gratitude. It is your suggestion that pushed the performance boundary of our work. The corresponding results have been updated in **Table 5** of the manuscript, and we have also updated the main results in **Table 1** and **Table 2** to reflect this stronger default.

---

> ### Author Response · Authors · 2025-11-18
>
> ## **Q2: Generalizability to Qwen2.5-VL and Comparison to "Thinking with Image" Methods**
>
> We thank the reviewer for this valuable comment. We fully agree that providing results on more recent and powerful MLLMs like Qwen2.5-VL is crucial for validating the generality and effectiveness of our proposed method.
> To address this, we have now conducted extensive experiments on the **Qwen2.5-VL 3B and 7B** baselines. We have also taken this opportunity to incorporate comparisons with recent RL-based "thinking with image" methods (e.g., DeepEyes[1] and Thyme[2]), which also build upon the Qwen2.5-VL architecture.
>
> As Qwen2.5-VL natively supports dynamic resolutions, we standardized the input for consistency. On most benchmarks, we set the number of visual tokens to 576 to align with the LLaVA and DeepSeek-VL baselines. For high-resolution benchmarks (V\* and HR-Bench), we increased this to a maximum of 4096 visual tokens to effectively process the detailed inputs and facilitate a fair comparison. The complete results have been added to **Table 1** (Document & OCR) and **Table 2** (Vision-Centric & High-Resolution) in the revised paper and are as follows:
>
>
> **Performance on Document & OCR benchmarks upon Qwen2.5-VL baseline.**
>
> | Methods | Throughput | DocVQA val  | ChartQA test  | OCRBench test  | InfoVQA val  | TextVQA val  | Ave. |
> | :--- | :---: | :---: | :---: | :---: | :---: | :---: | :---: |
> | **Qwen2.5-VL-3B** | 1.0x | 87.1 | 82.6 | 73.8 | 62.7 | 76.9 | 76.6 |
> | &nbsp;&nbsp;&nbsp;+SD-RPN(Ours) | 0.56x | **89.5**&uparrow;2.4 | **83.8**&uparrow;1.2 | **76.3**&uparrow;2.5 | **67.0**&uparrow;4.3 | **79.7**&uparrow;2.8 | **79.3**&uparrow;2.7 |
> | **Qwen2.5-VL-7B** | 1.0x | 92.0 | 82.6 | 81.5 | 70.2 | 81.1 | 81.5 |
> | &nbsp;&nbsp;&nbsp;+SD-RPN(Ours) | 0.57x | **93.6**&uparrow;1.6 | **85.5**&uparrow;2.9 | **82.9**&uparrow;1.4 | **76.9**&uparrow;6.7 | **83.5**&uparrow;2.4 | **84.5**&uparrow;3.0 |
>
> **Performance on Vision-Centric & High-resolution benchmarks upon Qwen2.5-VL baseline.**
>
>
> | Methods | Attr  (V*)  | Spatial  (V*)  | Overall  (V*)  | F1  (POPE)  | Acc.  (POPE)  | FSP  (4K)  | FCP  (4K)  | Overall  (4K)  | FSP  (8K)  | FCP  (8K)  | Overall  (8K)  |
> | :--- | :---: | :---: | :---: | :---: | :---: | :---: | :---: | :---: | :---: | :---: | :---: |
> | **Qwen2.5-VL-3B** | 81.7 | 60.5 | 73.3 | 87.8 | 88.6 | 80.5 | 52.0 | 66.3 | 70.0 | 47.8 | 58.9 |
> | &nbsp;&nbsp;&nbsp;+SD-RPN(Ours) | **91.3** | **65.8** | **81.2**&uparrow;7.9 | **88.9**&uparrow;1.1 | **89.3**&uparrow;0.7 | **88.0** | **58.3** | **73.1**&uparrow;6.8 | **79.5** | **51.0** | **65.3**&uparrow;6.4 |
> | **Qwen2.5-VL-7B** | 80.0 | 75.0 | 78.0 | 86.7 | 87.9 | 81.8 | 62.8 | 72.5 | 72.8 | 54.5 | 63.6 |
> | &nbsp;&nbsp;&nbsp;+Mini-o3 &dagger; | - | - | 88.2 | - | - | - | - | 77.5 | - | - | 73.3 |
> | &nbsp;&nbsp;&nbsp;+Thyme &dagger; | 83.5 | 80.3 | 82.2 | - | - | 91.0 | 63.0 | 77.0 | 86.5 | 57.5 | 72.0 |
> | &nbsp;&nbsp;&nbsp;+DeepEyes &dagger; | 91.3 | **88.2** | **90.1** | 87.7 | - | 91.3 | 59.0 | 75.1 | **86.6** | 58.5 | 72.6 |
> | &nbsp;&nbsp;&nbsp;+SD-RPN(Ours) | **96.5** | 79.0 | 89.5&uparrow;11.5 | **88.4**&uparrow;1.7 | **88.9**&uparrow;1.0 | **92.5** | **64.5** | **78.5**&uparrow;6.0 | 86.5 | **60.5** | **73.5**&uparrow;9.9 |
>
>
>
> As these new results show:
> On **Document & OCR benchmarks**, SD-RPN consistently boosts the average accuracy by approximately **3%** for the powerful Qwen2.5-VL series (e.g., boosting the 7B model from 81.5% to 84.5%).
> The improvement on **High-Resolution benchmarks** is even more significant, as resolution remains a primary obstacle in this scenario. Concretely, SD-RPN boosts the Qwen2.5-VL 7B baseline by **11.5% on V\***, **6.0% on HR-Bench 4K**, and **9.9% on HR-Bench 8K**.
> These results are **on par with or outperform** recent RL-based "thinking with image" methods like DeepEyes[1] and Thyme[2]. Notably, those methods often incur extra latency for their thinking mechanism (e.g., visual Chain-of-Thought), which can significantly impair efficiency.

---

> ### Author Response · Authors · 2025-11-18
>
> ## **Q3: Detailed Latency and GPU Cost Analysis**
>
> Thank you for raising this practical and important concern. In response to this comment, we have collected additional experimental data to quantify the efficiency improvements in detail. We now report the wall-clock latency (Total Time), GPU memory cost, and accuracy in the tables below.
>
> The first table compares methods using the **LLaVA-1.5** baseline across its 7B and 13B sizes. This analysis was conducted on the V* benchmark (191 high-resolution samples). (Please note: because these metrics are from a specific 191-sample run, the total time may not perfectly align with the relative throughput figures in Table 1, which were averaged across five different benchmarks.)
> The second table compares the **Qwen2.5-VL-7B** baseline with our method (+Ours) across a range of input image token counts. All metrics were measured on a single NVIDIA A6000 GPU, and accuracy was evaluated on the V* benchmark.
>
>
> **Detailed efficiency and accuracy analysis on LLaVA-1.5 on V\* Bench.**
> | V* (191 Samples) | Total Time (s) | GPU Memory (G) | Accuracy (%) |
> | :--- | :--- | :--- | :--- |
> | **LLaVA-1.5-7B**| 44 | 14.8 | 50.3 |
> | +S 2 | 70 | 14.8 | 55.5 |
> | +ViCrop | 124 | 21.8 | 52.4 |
> | +Ours | 73 | 16.9 | 67.5 |
> | **LLaVA-1.5-13B** | 65 | 27.0 | 50.8 |
> | +S 2 | 92 | 27.0 | 49.7 |
> | +ViCrop | 197 | 37.4 | 51.8 |
> | +Ours | 109 | 30.2 | 62.3 |
>
> **Detailed efficiency and accuracy analysis on Qwen2.5-VL-7B on V\* Bench. Each data point corresponds to a different maximum number of visual tokens.**
>
> | Methods | Metric | 128 | 256 | 576 | 1024 | 1536 | 2048 | 3072 | 4096 |
> | :--- | :--- | :---: | :---: | :---: | :---: | :---: | :---: | :---: | :---: |
> | **Qwen2.5-VL-7B** | Total Time (s) | 21.5 | 31.4 | 54.4 | 92.0 | 143.9 | 206.9 | 358.2 | 526.6 |
> | | Output Tokens/s | 21.1 | 15.5 | 9.1 | 5.4 | 3.4 | 2.4 | 1.4 | 1.0 |
> | | GPU Memory (G) | 16.4 | 16.6 | 17.2 | 17.5 | 19.1 | 19.6 | 20.8 | 23.5 |
> | | Accuracy (%) | 42.9 | 53.4 | 64.4 | 70.2 | 72.8 | 72.3 | 74.9 | 78.0 |
> | **+Ours** | Total Time (s) | 36.7 | 57.0 | 107.8 | 146.6 | 196.2 | 251.7 | 410.4 | 568.1 |
> | | Output Tokens/s | 13.0 | 8.6 | 5.1 | 3.0 | 2.5 | 2.1 | 1.3 | 0.9 |
> | | GPU Memory (G) | 18.8 | 19.2 | 20.1 | 20.7 | 22.2 | 22.9 | 24.8 | 26.5 |
> | | Accuracy (%) | 59.7 | 70.7 | 77.5 | 85.9 | 86.4 | 83.2 | 89.5 | 87.9 |
>
> We have included this important data and a detailed discussion in the revised paper's **Appendix C.1**, which includes **Figure 6** for an intuitive visualization of the performance-throughput trade-off.
> This new analysis clearly demonstrates the superiority of our method's trade-off. Notably, on Qwen2.5-VL-7B, our method achieves **77.5% accuracy at 5.1 tokens/s** (with a 576-token input). The baseline model only reaches a similar performance level (78.0%) at a 4096-token input, where its throughput is a much lower **1.0 tokens/s**. We wish to thank you once again for this constructive comment, which has helped us significantly strengthen the paper's efficiency analysis.

---

> ### Author Response · Authors · 2025-11-18
>
> ## **Q4: Ground-Truth Bounding Box Performance**
>
> We deeply appreciate the reviewer for this thoughtful and insightful question.
> Our original training dataset, sourced from LLaVA-1.5, did not include ground-truth (GT) bounding box annotations. Motivated by the reviewer's comment, we conducted a new experiment using the VisualCoT[3] dataset, which provides question-relevant bounding boxes. We sampled 68K QA pairs (18K from TextVQA, 50K from GQA) to create a new training set, '68k-VCoT', covering both text-rich and natural images.
> We then trained two versions of our SD-RPN on this set:
> 1.  **Pseudo-label (Ours):** Trained using our proposed self-distillation pipeline.
> 2.  **GT Box:** Trained using the GT bounding boxes. To use these, we converted the boxes into dense labels, marking all tokens *inside* the box as foreground (1) and all tokens *outside* as background (0).
>
> The results on the LLaVA-1.5-7B baseline are summarized below:
>
> **Comparison of SD-RPN trained with pseudo-labels vs. Ground-Truth (GT) Bounding Boxes on LLaVA-1.5-7B Baseline**
>
> | Method | Annotation | OCRBench | TextVQA | POPE | V\* | Ave. |
> | :--- | :--- | :---: | :---: | :---: | :---: | :---: |
> | Baseline | - | 31.4 | 46.1 | 85.9 | 50.3 | 53.4 |
> | 68k-VCoT | Pseudo-label | **36.9** | 58.3 | **88.0** | **66.0** | **62.3** |
> | 68k-VCoT | GT Box | 36.1 | **59.3** | 87.5 | 60.7 | 60.9 |
>
> The results are revealing. The 'GT Box' version performs better on TextVQA (+1.0%), which is an **in-domain** dataset (since 18K training samples came from TextVQA). However, our 'Pseudo-label' version generalizes significantly better to **unseen** benchmarks, outperforming the GT version on OCRBench, POPE, and most notably by **over 5% on the V\* benchmark**.
> We attribute this counter-intuitive result to two primary factors, which align with our paper's core hypotheses:
>
> 1.  **Internal Consistency:** As discussed in **Section 3.3**, attention maps from the MLLM's *own* responses provide a more internally consistent and attainable distillation target. The 'Pseudo-label' version converges faster as the RPN is learning to predict a signal (denoised attention) that is native to the MLLM's grounding mechanism. The 'GT Box' version, trained on external annotations, converges more slowly and may be under-trained with this limited 68K dataset.
>
> 2.  **Label Quality:** Our pseudo-label pipeline is designed to be *sparse*, explicitly ignoring ambiguous tokens. In contrast, a GT *bounding box* is a *dense and crude* approximation. For objects with irregular shapes, a box erroneously labels background tokens *inside* the box as foreground. This introduces significant noise, forcing the RPN to learn to predict overly large or inaccurate regions, which hinders performance on fine-grained perception tasks (as seen in the V\* results).
>
> Therefore, we conclude that when the training set is large enough, or when the training and testing set is in the same domain, the GT version will definitely be the upper bound of SD-RPN. However, when the training set is limited, it may lag behind SD-RPN due to limited generality, as our experimental results demonstrate.

---

> ### Author Response · Authors · 2025-11-18
>
> ## **Q5: Rationale for BCE Loss over Alternatives (e.g., Focal Loss)**
>
> We thank the reviewer for this insightful question regarding our choice of loss function.
> In fact, we had experimented with Focal Loss in the early stages of our work and found it performed slightly worse than the BCE version. In response to this valuable question, we have now complemented our paper with a direct ablation study to compare these loss functions on the LLaVA-1.5-7B baseline.
>
> **Ablation study on loss functions.**
>
> | | OCRBench | TextVQA | POPE | V\* | Ave. |
> | :--- | :---: | :---: | :---: | :---: | :---: |
> | Baseline | 31.4 | 46.1 | 85.9 | 50.1 | 53.4 |
> | **BCE** | 37.0 | 58.7 | 87.1 | 67.5 | **62.6** |
> | **FocalLoss** | 36.4 | 58.2 | 87.1 | 63.5 | 61.3 |
>
> As the results show, Focal Loss lags behind the naive BCE by 1.3% on the average score. We attribute this counter-intuitive result to the key difference between *clean labels* and our *noisy pseudo-labels*.
> We hypothesize that Focal Loss, by design, emphasizes "hard" positive samples. When labels are noisy (as ours are, being derived from attention), this mechanism may inadvertently cause the model to become *more sensitive* to and overfit on these noisy pseudo-positive tokens. In our qualitative visualizations, we also found that the RPN trained with Focal Loss tended to predict a noisier RoI map with more false foreground activations compared to the BCE version.
>
> Regarding IoU-based losses, these are difficult to apply directly to our framework. Our RPN is designed to predict a *dense, 2D RoI map* (a score for every token), not a sparse bounding box. This architectural choice makes a box-based IoU loss inapplicable. Given this, the standard BCE loss proved to be a more stable and highest-performing choice for our specific task of learning from self-distilled, noisy attention maps.
>
> [1]. Zheng, Ziwei, et al. "DeepEyes: Incentivizing" Thinking with Images" via Reinforcement Learning." arXiv:2505.14362.
>
> [2]. Zhang, Yi-Fan, et al. "Thyme: Think beyond images." arXiv:2508.11630.
>
> [3]. Shao, Hao, et al. "Visual cot: Advancing multi-modal language models with a comprehensive dataset and benchmark for chain-of-thought reasoning." NeurIPS (2024).

---

> ### Public Comment · ~Sibo_Yin2 · 2025-11-18
>
> I believe the authors have already conducted experiments on Qwen2.5-VL, as shown in Table 1.

---

> ### Author Response · Authors · 2025-11-25
>
> Dear Reviewer 2wEb,
>
> We are writing to respectfully invite you to review our responses and the new experimental results we have provided.
>
> As the author-reviewer discussion phase is concluding soon, we are keen to ensure that our revisions have adequately addressed your concerns. We would greatly appreciate any further feedback or confirmation you might have before the deadline.
>
> Thank you again for your time and valuable suggestions.
>
> Best regards, The Authors

---

> > ### Comment · Reviewer_ScXn · 2025-11-27
> >
> > Thanks the authors for the responses. My concerns are mostly addressed. I will take this into account when making my final decision.

---

> > > ### Author Response · Authors · 2025-11-27
> > >
> > > Dear Reviewer ScXn,
> > >
> > > Thank you for your prompt response.
> > >
> > > We are glad to hear that our revisions and clarifications have adequately addressed your concerns. We truly appreciate your valuable feedback throughout this process; your recognition and support motivate us to conduct deeper exploration in this direction.
> > >
> > > We wish you all the best.
> > >
> > > Sincerely, The Authors

---

### Author Response · Authors · 2025-11-18

Dear Reviewers,

We sincerely thank you for your insightful comments and deeply appreciate the time and effort you dedicated to thoroughly reviewing our manuscript.

Your constructive feedback has been invaluable. It has significantly improved the rigor and clarity of our paper and motivated us to conduct deeper explorations into the topic.

We have made every effort to address each of your questions, and we hope our rebuttal adequately addresses your concerns. We have incorporated your valuable suggestions into the revised version of our paper. The newly revised contents are highlighted in blue to distinguish them from the original script.

Thank you once again for your significant contribution to our research community.

Best Regards,

Paper379 Authors.

---

### Author Response · Authors · 2025-12-01
**Summary of Rebuttal for the Area Chair**

**Dear Area Chair,**

We sincerely thank you for your time and effort in managing the review process under these exceptional circumstances. We also wish to express our deep gratitude to the reviewers for their constructive feedback and their continued engagement, which has significantly strengthened this manuscript. We understand the workload involved in the recent "reset," and we hope this summary assists you in your assessment.

During the discussion phase, we engaged in extensive dialogue with all reviewers. We are pleased to report that **Reviewer jaiz** and **Reviewer KUum** explicitly confirmed our rebuttal resolved their concerns (with **Reviewer jaiz maintaining a Score of 8: Accept**). **Reviewer ScXn** also noted concerns were "mostly addressed."

Below is a summary of the key actions we took to strengthen the paper:

### 1. Significant Generalization to Stronger Baselines (Qwen2.5-VL)
* **Addressed concern from:** Reviewer **2wEb** (Primary), also relevant to **ScXn** and **jaiz**.
* **Action:** We extended our method to the state-of-the-art **Qwen2.5-VL (3B and 7B)**.
* **Result:** SD-RPN yields consistent improvements on this powerful baseline. Notably, on High-Resolution benchmarks, we boost **Qwen2.5-VL-7B** by **+11.5% on V\***, **+6.0% on HR-Bench 4K**, and **+9.9% on HR-Bench 8K**. We also outperformed recent "Thinking with Image" methods (e.g., DeepEyes, Thyme).

### 2. Rigorous Ablation Studies (Thresholds, Loss & Layers)
* **Addressed concern from:** Reviewers **2wEb**, **KUum**, and **jaiz**.
* **Action:** We conducted comprehensive ablations on the pseudo-label thresholds ($\tau_{fg}, \tau_{bg}$), loss functions (BCE vs. Focal Loss), and RPN layer depth.
* **Result:** This investigation identified a new optimal threshold configuration ($\tau_{fg}=0.20, \tau_{bg}=0.05$) which improves average performance by nearly **1%** over our initial submission. We have updated **Tables 1, 2, and 5** to reflect these stronger results.

### 3. Detailed Efficiency & Cost Analysis
* **Addressed concern from:** Reviewers **2wEb**, **ScXn**, and **jaiz**.
* **Action:** We provided a detailed breakdown of wall-clock latency, GPU memory, and throughput in response to reivewer and **Appendix C.1**. We also clarified the "lightweight" claim as requested by Reviewer jaiz.
* **Result:** We demonstrated that SD-RPN is **20x faster** than external grounding models (e.g., Grounding-DINO) because it re-uses MLLM features, while simultaneously achieving superior performance.

### 4. Comparison with SoTA & Ground Truth Validation
* **Addressed concern from:** Reviewers **jaiz** (SoTA), **2wEb** (GT Boxes), and **ScXn** (Justification).
* **Action:** We added comparisons to SoTA High-Res methods (Monkey, LLaVA-NeXT, etc.) in **Appendix C.4**. We also experimented with training on Ground-Truth (GT) bounding boxes vs. our Pseudo-labels.
* **Result:** Our "Pseudo-label" approach generalizes significantly better to unseen benchmarks (outperforming GT-trained models by **5% on V\***), validating our hypothesis that learning from the model's internal attention is superior to forcing alignment with external, crude bounding boxes.

### 5. Qualitative Analysis (Noise & Failures)
* **Addressed concern from:** Reviewers **KUum** and **ScXn**.
* **Action:** We added **Figure 7** (analyzing attention noise sources) and **Figure 9** (detailed failure case analysis) to the appendix.
* **Result:** We provided a transparent analysis of when and why the method succeeds or fails, offering deeper insights into the behavior of self-distillation.

---

### Reviewer Consensus (Prior to Reset)

We successfully addressed the vast majority of questions. The reviewers' final comments before the reset are summarized below:

* **Reviewer jaiz (Score: 8 - Accept):**
    > *"My questions have been clearly addressed... With the added experiments, revised phrasing and detailed explanation, I believe the updated paper can contribute meaningfully to the research community in VLM. Therefore, I keep the score of 8, and support the acceptance of this paper."*
* **Reviewer KUum (Score: 6 - Positive score maintained):**
    > *"Thank you for the detailed response. The rebuttal has successfully clarified the points I raised. I will maintain my positive score."*
* **Reviewer ScXn (Score: 4 - Concerns are mostly addressed):**
    > *"Thanks the authors for the responses. My concerns are mostly addressed. I will take this into account when making my final decision."*
* **Reviewer 2wEb (Score: 4 - No response):**
    * *Status:* No official response received yet. However, we have fully addressed primary request for **Qwen2.5-VL experiments**, **ablation studies**, and **cost analysis** in the revised manuscript.

We thank you again for your time and dedication to the review process.

**Best regards,**

**Paper 379 Authors**

---

### Meta-Review · Area_Chair_54hM · 2026-01-06

**Summary:**

The paper proposes the SD-RPN to enhance fine-grained perception in MLLMs without requiring expensive human annotations or heavy computation. The reviewers generally appreciated the novelty of the self-distillation approach, which leverages the model's internal attention maps. The initial decision was informed by several concerns regarding the rigorousness of the evaluation. Reviewers questioned the method's performance relative to sota high-resolution methods and its generalizability to stronger, more recent backbones like Qwen2.5-VL. There were also requests for deeper ablations regarding hyperparameters (specifically thresholds for pseudo-labeling), loss functions, and a more concrete analysis of computational efficiency compared to external object detectors. Finally, reviewers sought clarification on why this method is superior to using ground-truth bounding boxes or external grounding tools.

Based on the overall evaluation, I recommend accepting this paper.

**Reviewer Concerns:**

Addressed Concerns:

The authors provided a comprehensive and highly effective rebuttal that addressed the vast majority of reviewer concerns:
Generalizability to Stronger Models (Reviewer 2wEb): The authors successfully integrated their method with the state-of-the-art Qwen2.5-VL (3B and 7B). The results showed significant improvements (e.g., +11.5% on the V* benchmark), demonstrating that the method scales well with stronger base models.

Comparison with SoTA (Reviewer jaiz): The authors added detailed comparisons with high-resolution methods like Monkey, LLaVA-NeXT, and Mini-Gemini, contextualizing their performance and highlighting the data-efficiency of their approach.

Ablation Studies (Reviewers KUum, 2wEb): The authors conducted the requested ablations on pseudo-label thresholds and loss functions (BCE vs. Focal Loss). This not only justified their design choices but actually led to a new configuration that improved performance by ~1%.

Efficiency Analysis (Reviewers jaiz, ScXn): A detailed breakdown of wall-clock latency, GPU memory, and throughput was provided, demonstrating that SD-RPN is significantly faster (~20x) than using external models like Grounding-DINO.

Method Justification (Reviewer ScXn): The authors provided empirical evidence showing that their self-distillation approach generalizes better to unseen benchmarks than models trained on Ground-Truth bounding boxes, validating the hypothesis that learning from the model's internal attention is more consistent for MLLMs.

Qualitative Analysis (Reviewer KUum): The authors added visualizations of attention noise and a detailed failure case analysis.

Outstanding Concerns:

No significant technical concerns remain. Reviewer jaiz and KUum explicitly confirmed their satisfaction. Reviewer ScXn noted concerns were "mostly addressed." Reviewer 2wEb did not respond post-rebuttal, but the specific experiments they requested (Qwen2.5-VL) were performed with positive results.

**Reviewer Scores:**

Reviewer jaiz: Originally 8. The reviewer explicitly maintained this score post-rebuttal.

Reviewer KUum: Originally 6. The reviewer explicitly maintained a positive score post-rebuttal.

Reviewer ScXn: Originally 4. The reviewer acknowledged concerns were "mostly addressed," suggesting a likely shift to a weak accept.

Reviewer 2wEb: Originally 4. This reviewer's primary critique was the lack of Qwen2.5-VL experiments. Since the authors added these with strong results, it is highly probable the reviewer would have raised their score significantly had they participated in the discussion.

---

### Decision · Program_Chairs · 2026-01-26

Accept (Poster)